# Benthic Foraminifera Diversity of the Abyssal Northwest Atlantic

Michael Hesemann [1,2] 

1 Senckenberg am Meer, German Centre for Marine Biodiversity Research (DZMB), Südstrand 44, 26382 Wilhelmshaven, Germany; michael.hesemann@senckenberg.de or hesemann@foraminifera.eu
2 The Foraminifera.eu Lab, Waterloostr. 24, 22769 Hamburg, Germany

**Abstract:** The species diversity of benthic foraminifera at four abyssal working areas in the Labrador Sea, Labrador Basin, and Southwest of the Azores is documented. One hundred and fifty taxa (forty-three not assigned to a species) were found and their diversity was recorded. One hundred and twenty-four taxa (fifteen not assigned to a species) were illustrated with optical and/or SEM (Scanning Electron Microscope) images on twelve plates. The material was sampled during RV Sonne cruise SO286 as part of IceDivA2 (Icelandic marine Animals meets Diversity along latitudinal gradients in the deep sea of the Atlantic Ocean 2). IceDivA2 investigated the biodiversity within key groups of the marine benthic abyssal habitats of the North Atlantic. Thirty-two samples from four sliced and three full cores, from six stations sampled with a MUC (Multiple corer), were analyzed. Given low sedimentation rates in such environments the material is assumed to be of Holocene to late Pleistocene age. Due to the scarcity of living specimens this study was based on total assemblages. Four species-based clusters are identified, which correspond to the four working areas. The samples of each cluster show specific characteristics markedly different from those of the samples of the other clusters. It indicates that abyssal faunas are heterogeneous. Three clusters are dominated by *Epistominella exigua* (Brady, 1884), which is recorded as not rare to dominant in many abyssal plains worldwide. The faunal differences are manifested in the long tail of less important species and differing abundances of *E. exigua*.

**Keywords:** benthic foraminifera; biodiversity; abyssal Northwest Atlantic; Labrador Sea; Labrador Basin

## 1. Introduction

### 1.1. Importance of Abyssal Ecosystems and Their Foraminifera

More than 50% of the surface of the Earth is covered by abyssal ocean deeper than 2000 m. Long regarded as a desert-like ecosystem it is known since the expeditions by HMS Lightning, HMS Porcupine, and HMS Challenger from 1869 to 1876 that the deep sea is a highly diverse habitat. To date it is still largely unexplored [1]. The IceDivA2 expedition SO286 was launched to record deep sea marine biodiversity west of the Mid-Atlantic Ridge [2]. Benthic foraminifera are an important component of the benthos with increasing water depth and may account for 50% of the biomass on abyssal plains [3]. In this study the biodiversity of foraminifera at six stations of the abyssal northwestern Atlantic was recorded and investigated.

There is no universal agreement on depth divisions in the oceans, even in the field of foraminifera. Some authors view habitats at depths below 4000 m as abyssal [3,4], while others draw the line between bathyal and abyssal at 2000 m [5,6]. In this study we refer to depths which the samples were from as abyssal (2508 m to 3685 m).

### 1.2. What Are Foraminifera?

Foraminifera are single-celled, eukaryote organisms bearing a shell and are described as amoeboid protists. They build one or more chambers. Their name refers to the openings

(Latin: foramen), which connect the chambers and to the external environment. While important parts of the cell and plasma are protected within the shell, plasma and plasma strings extend outside of the shell to catch food. Foraminifera are secondary producers and feed on organic matter with a wide range of strategies. Besides fresh water habitats they are abundant in all marine environments from the upper intertidal zone to deep ocean trenches. Planktonic foraminifera drift in open marine environments at different water depths. Benthic foraminifera inhabit the seafloor from hard to soft grounds, and from positions on top of the water–debris interval to several centimeters within the sediment. In the marine realm, foraminifera build robust shells, which if not dissolved or crushed accumulate in the sediment and leave a fossil record [4,7]. More than 49,000 fossil and recent species are described, of which almost 9000 are recorded as recent [8]. The identification of taxa is almost exclusively based on morphological features such as wall material and chamber arrangement. DNA data are only known for a small number of recent species and their consideration in taxonomy is still being debated [9]. This study is based exclusively on the morphological approach. To clarify the identifications 124 taxa are illustrated on Appendix A Figures A1–A12.

### 1.3. Previous Studies

The foraminiferal biodiversities in the APLA (=abyssal plains of the Labrador Sea, Labrador Basin, and until latitude 35° N) are relatively unexplored. In Brady's classic HMS challenger report from 1884 [10] the presence of 24 benthic taxa is reported from abyssal stations 64 and 70 southwest of the Azores. Cushman's eight part series on Foraminifera of the Atlantic [11–18] based partly on Flint [19] lack any notion as the APLA were not sampled. In 1953 Phleger et al. reported modern and near modern full foraminiferal assemblages from the North Atlantic in cores of 0.5 m to 19.5 m lengths. It included seven stations from west of the Azores ranging from water depths of 2640 m to 4940 m and from one station at the eastern limit of the Labrador Basin at 47°24′ N, 30°03′ W from 3383 m depth [20]. They focused on planktonic taxa and report that the scarcity of benthic taxa hindered a detailed interpretation. Nonetheless they gave for each of the eight stations counting data for 16 to 20 taxa of the surface layers. They also made drawings of the taxa. They concluded that most deep water benthics are geographically widespread in the oceans reflecting uniform conditions in the deep. 1974 Schnitker characterized faunas in the whole abyssal West Atlantic between the latitudes 35° N and 65° N as *Epistominella exigua* (Brady, 1884) dominated and connected it with Arctic Bottom Waters [21,22]. For the APLA his findings were based on samples from 14 stations and none in the Labrador Sea. A species distribution list was not given. Distribution data on benthic foraminifera for four stations of the beginning of the APLA at the Nova Scotia rise were given in a study of arenaceous foraminifera in the Northwest Atlantic by Schröder from 1986 [23]. The APLA assemblages were characterized as *E. exigua* dominated and associated with Antarctic Bottom Water coming from the south. The arenaceous taxa were illustrated. Bilodeau et al. [24] used 13 foraminiferal taxa for their study of water mass changes in the Labrador Sea and Irminger Basin. The seven stations 6–9 and 14–16 are in the APLA. At five stations the surface samples were dominated by *E. exigua/Oridorsalis umbonatus* (Reuss, 1851), at one by *Nuttallides umbonifer* (Cushman, 1933), and at another by *Cibicidoides wuellerstorfi* (Schwager, 1866). The dominance of *E. exigua* was interpreted as indicative of North East Atlantic Deep Water and that of *C. wuellerstorfi* as indicative of North West Atlantic Deep Water. Illustrations of 42 taxa from six stations in the APLA were given in an image dataset from 2019 [25]. A list of taxa and distribution data per station were not given.

### 1.4. Focus of This Study

So far comprehensive distribution data for the APLA, spreading over more than 2.5 million km², are only given for four stations. This study documents the benthic foraminiferal biodiversities from six more stations (Table 1, Supplementary Materials Tables S1 and S2). Distribution lists of all taxa are given per sample and one hundred and twenty four taxa

are illustrated (Supplementary Materials Tables S1 and S2, Appendix A Figures A1–A12). Cluster analysis was applied to reveal the faunal similarities and distances between samples and between stations.

**Table 1.** Details for the stations of RV Sonne cruise SO286 where the investigated material was sampled.

| Station | Working Area | Longitude | Latitude | Depth | Device | Sediment | Facies |
|---------|-------------|-----------|----------|-------|--------|----------|--------|
| SO286_21 | Labrador Sea | 58°14.422′ | 54°13.075′ | 3391 m | MUC | soft mud | globigerina ooze |
| SO286_42 | Labrador Basin | 51°58.255′ | 38°59.534′ | 3685 m | MUC | soft mud | globigerina ooze |
| SO286_65 | Azores SW | 37°00.025′ | 35°29.491′ | 3193 m | MUC | soft mud | globigerina ooze |
| SO286_66 | Azores SW | 37°00.032′ | 35°29.441′ | 3192 m | MUC | soft mud | globigerina ooze |
| SO286_67 | Azores SW | 37°00.037′ | 35°29.382′ | 3209 m | MUC | soft mud | globigerina ooze |
| SO286_75 | Seamount | 37°13.922′ | 35°32.316′ | 2508 m | MUC | soft mud | globigerina ooze |

## 2. Materials and Methods

### 2.1. Materials

The investigated material consisted of thirty-two samples from seven cores (Appendix A Table A1), which were collected with a multicorer (MUC) at six stations (Table 1) in the APLA during the expedition SO286 of RV Sonne in November–December 2021 [1]. The MUC tube had a diameter of 10 cm.

The water depths of the stations range from 2508 m to 3685 m. The sampling was done in four working areas: Area Labrador Sea with stations SO286_21; Area Labrador Basin with station SO286_42; Area Azores SW with stations SO286_65, _66, and _67 and Area Seamount with station SO286_75. A map with the stations is given in Figure 1. The material is stored at the Foraminifera.eu Lab facility.

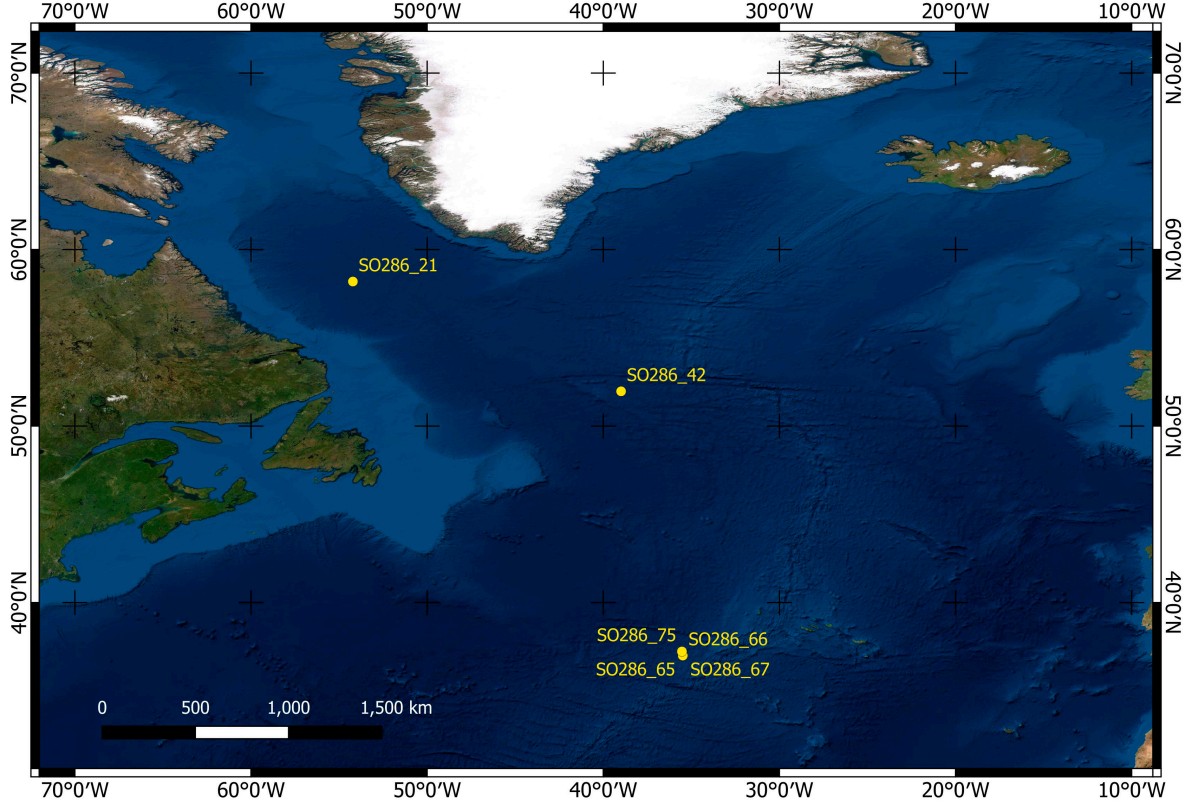

**Figure 1.** Map of the stations of RV Sonne cruise SO286 where the investigated material was sampled.

*2.2. Methods*

On board, three of the MUC-cores were packed in full in labeled plastic bags and frozen at −20 °C. Four of the MUC-cores were cut from the sediment surface down to 10 cm depth in slices of 2 cm thickness, and then in slices of 5 cm thickness. A list of the samples is given in Appendix A Table A1. Each slice was packed in a labeled plastic bag and frozen at −20 °C. At this temperature it was stored and transported to the Foraminifera.eu lab and stored there at −18 °C. Each material packed in a single plastic bag, either being a full core or a slice, is referred to as sample. The densities of foraminifera (amount of specimens) per 100 g were calculated by weighing the frozen samples in total, the dried retained part, the washed-, dried part, the analyzed share, and the specimens count. The details are given in Appendix A Table A2. The not retained part of the sample was brought to room temperature of about 18 °C and stained for 24 h with rose Bengal dye, to differentiate between living and nonliving specimens. Then the sample was gently washed over a 125 µm meshed sieve, the residue was dried under cover for 7 days and split into fractions of between 0.1–0.2 g using a micro splitter. The chosen fraction >125 µm falls into the range commonly used for studies of recent foraminiferal assemblages [26]. It avoids misidentifications of smaller and juvenile specimens and facilitates the microscopic work. Pieces of broken specimens may have also been washed away through the 125 µm sieve. Their number was negligible. Specimens were not counted if more than 25% of test was missing. The Asa (Amount of specimens alive) was very low (Appendix A Table A2) and the study was based on total assemblages.

Sample SO286_67_08 was used to decide on the number of specimens to be counted in order to achieve an adequate assessment of the diversity. After counting 300 specimens in sample SO286_67_08 a total of 52 species were found. The 45 most abundant species were found after counting 150 specimens (Figure 2). They account for 98.0% of all specimens (Figure 3). This was seen as sufficient to assess diversity for the purposes of this study. Fractions were analyzed using a stereo microscope until at least 150 benthic specimens were found and identified to the species level. The last fraction was analyzed in full, which resulted in higher counts than 150. The counts and the derived aggregated data were stored and calculated in Microsoft Excel 2013.

The identification of the taxa was based upon Cushman [11–18], Schröder [23], Schmiedl [27], Stefanoudis [28], Tikhonova et al. [25], and Ellis & Messina [29]. The nomenclature was adjusted to that of the World Foraminifera Database [8].

For scanning electron microscope images, selected specimens were placed on stubs in the Foraminifera.eu Lab. They were coated with gold and imaged using a VEGA3-TESCAN scanning microscope (SEM) at Senckenberg am Meer. Photomicrographs of selected specimens were taken with a Keyence VHX 900F digital microscope in the Foraminifera.eu Lab.

Statistical parameters and the cluster analysis were calculated with PAST (PAlaeontological STatistics) data analysis package version 4.05 [30,31]. The species richness and diversity is based on the counts of specimens on the species level per sample (Supplementary Table S1). They were used to calculate the indices Fisher's alpha ($\alpha$), Pielou's equitability (J) and Shannon (H) [32–34]. The Fisher's alpha index ($\alpha$) describes the relationship between the number of species and the number of specimens by assuming that species abundance follows a log distribution [4]. It is a measure of diversity. Pielou's equitability (J) describes how specimens are divided between species and ranges between 0 and 1. It is a measure of evenness. The Shannon H index takes into account both the number of species and the evenness of distribution. For one specimen it is 0 and rises with the number of species and the evenness of the assemblage.

The assemblages were clustered using the hierarchical cluster analysis in Q-mode with Ward's method algorithm [35] based on the counts of specimens per species per sample (Supplementary Table S1). For faunal interpretations the counts were aggregated according to the type of wall material and life position. The wall material of the genera in this study was either organic, agglutinated, porcelaneous or calcareous as described in [36]. The life position was either epifaunal, infaunal, both, or unknown as described per species or genus in [4,37].

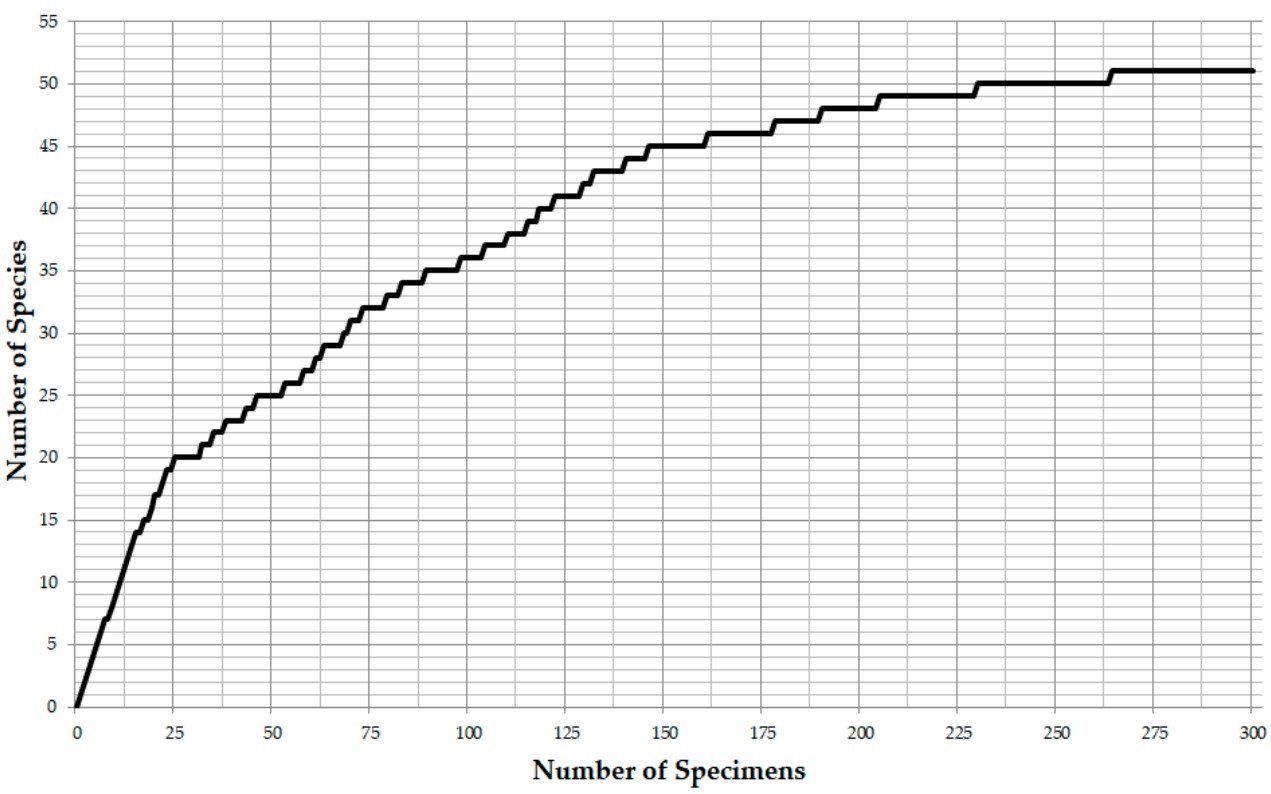

**Figure 2.** Relationship between number of counted specimens and number of species in sample SO286_67_08 of RV Sonne cruise SO286. Rarefaction Curve.

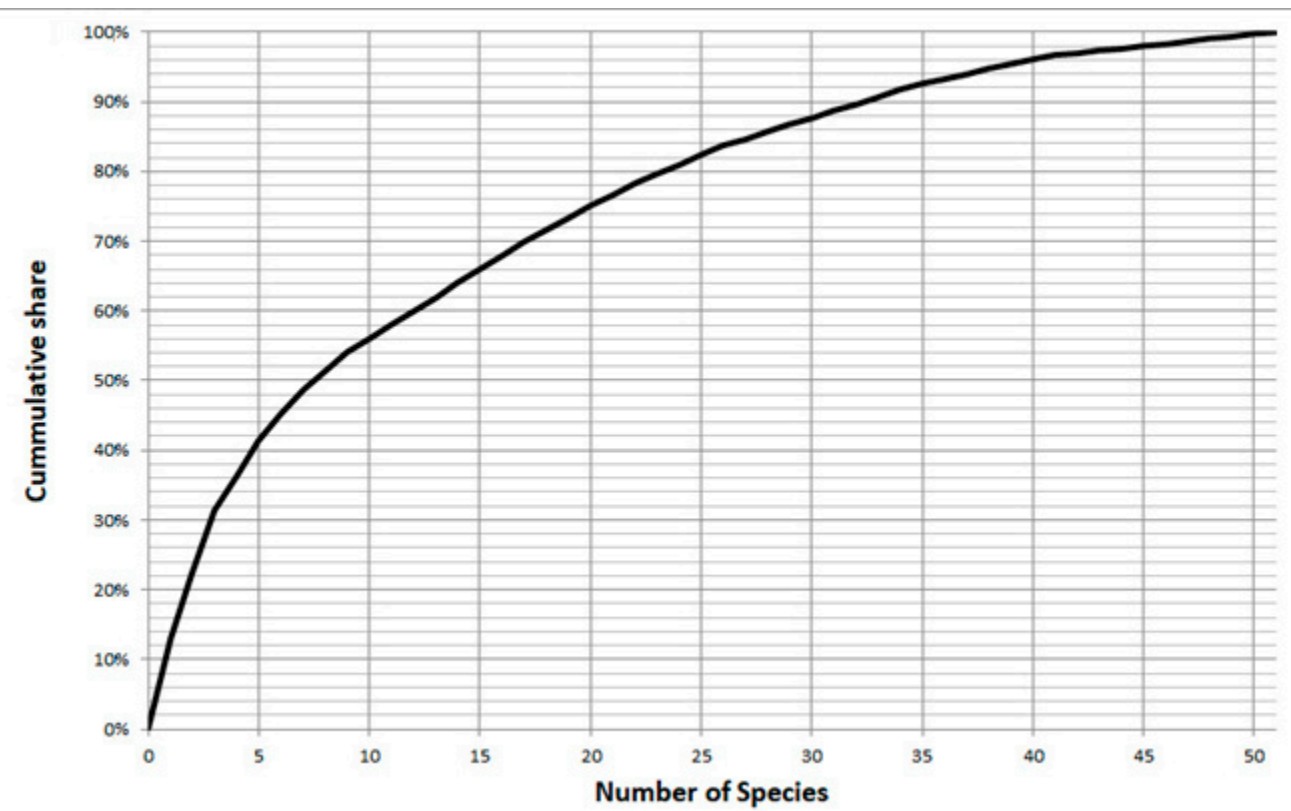

**Figure 3.** Relationship between numbers of species sorted by abundances and cumulative share of specimens in the assemblage in sample SO286_67_08 of RV Sonne cruise SO286.

### 3. Results

*3.1. Identified Taxa*

Based on the identification of 4986 benthic foraminiferal specimens from 32 samples to species level, a total of 150 foraminiferal species were found. These taxa belong to 12 orders and 92 genera (Table 2). The species list is given in Appendix A Table A3, the counts in Supplementary Table S1, and their shares in the samples in Supplementary Table S2.

**Table 2.** Benthic foraminiferal orders and number of genera and species recognized in samples from 6 stations of RV Sonne cruise SO286 in abyssal plains of the Labrador Sea, Labrador Basin, southwest of the Azores and at the flanks of a seamount southwest of the Azores.

| Order/Wall Material | Number of Genera | Number of Species |
|---|---|---|
| Astrorhizida | 6 | 7 |
| Hormosinida | 3 | 6 |
| Lituolida | 13 | 21 |
| Spirillinida (agglutinated) | 3 | 4 |
| Textulariida | 4 | 4 |
| Not assigned to a taxon | 1 | 1 |
| **Subtotal Agglutinated** | **30** | **43** |
| Miliolida | 12 | 23 |
| **Subtotal Porcellaneous** | **12** | **23** |
| Nodosariida | 3 | 10 |
| Polymorphinida | 9 | 19 |
| Robertinida | 2 | 2 |
| Rotaliida | 33 | 48 |
| Spirillinida (hyaline) | 1 | 2 |
| Vaginulinida | 1 | 3 |
| **Subtotal Hyaline** | **49** | **84** |
| Allogromiida | 1 | 1 |
| **Subtotal Organic** | **1** | **1** |
| **Total** | **92** | **150** |

*3.2. Identified Assemblages*

The hierarchical cluster analysis in Q-mode with Ward's method algorithm distinguished four main clusters, which correspond with the four working areas (Figure 4). Besides of SO286_75_25 and SO826_42_30 all samples of one core are clustered together. The clusters show specific characteristics in diversity, evenness, and species composition, which are discussed in the following text based on Table 3.

Cluster I corresponding with the Labrador Sea (Figure 4) is dominated by the species *Epistominella exigua* and *Oridorsalis umbonatus*. Further important species are *Hyperammina elongata* Brady, 1878, *Cibicidoides wuellerstorfi*, and *Pullenia quinqueloba* (Reuss, 1851) (Table 3, Supplementary Materials Table S2). The diversity and share of infaunal species are significantly lower than in the clusters southwest of the Azores and similar to those of cluster II. With 27% the share of agglutinates, it is 3 to 5 times higher as in the other clusters (Table 3). Cluster II corresponds with the Labrador Basin except for sample SO286_42_30 (Figure 4). It resembles cluster I in dominant species, low diversity, and low shares of infaunal species. The dominance of *Epistominella exigua* is more significant, which results in the lowest evenness of all clusters. The only further species of importance is *Guttulina communis* (d'Orbigny, 1826) (Table 3, Supplementary Materials Table S2).

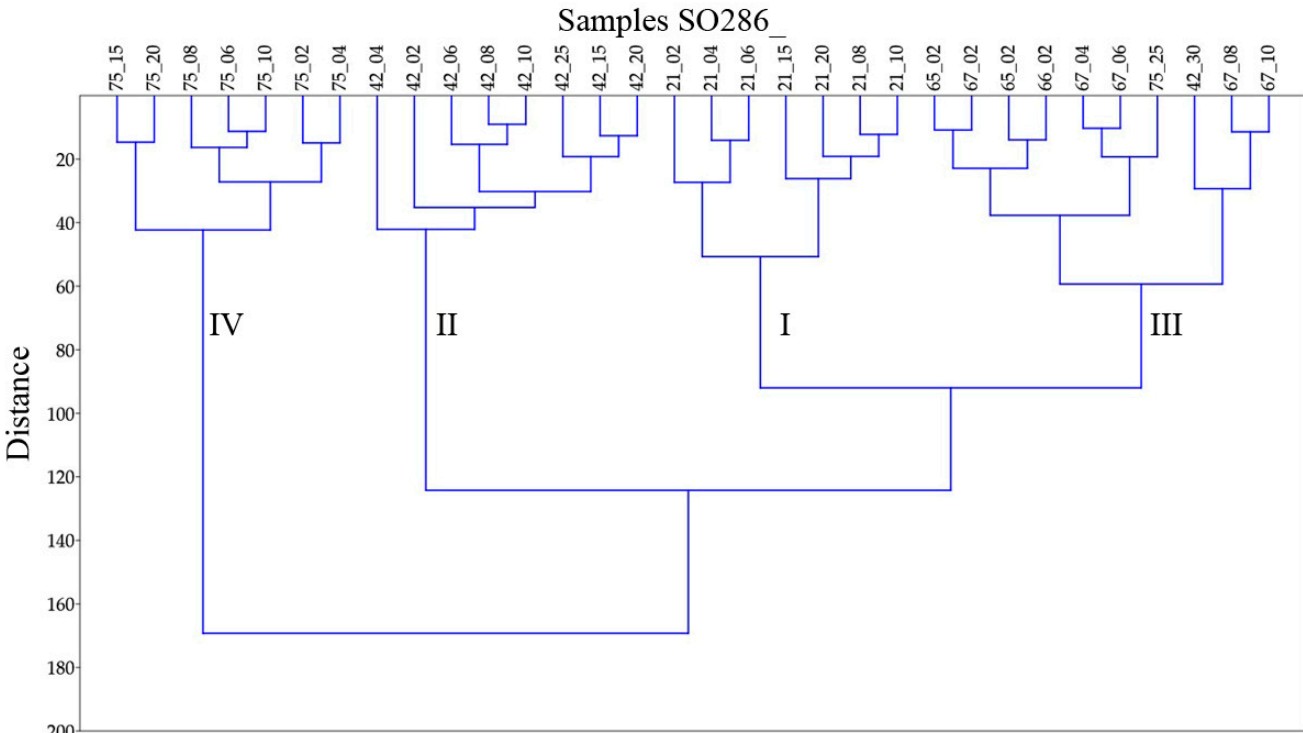

**Figure 4.** Dendrogram of benthic foraminiferal associations in 32 samples from abyssal plains in the Labrador Sea, Labrador Basin and southwest of the Azores collected during RV Sonne cruise SO286. The dendrogram is built using the hierarchical cluster analysis in Q-mode with Ward's method algorithm. It shows the four clusters I, II, III and IV.

Cluster III corresponds with the area southwest of the Azores with the exemption of samples SO286_42_30 and SO286_75_25 (Figure 4). It is dominated by the species *Epistominella exigua.* Further important species are *Cibicidoides wuellerstorfi* and the *Melonis* group (Table 3, Supplementary Materials Table S2). In comparison to clusters I and II the mean diversity measure α is more than 50% higher, and the mean shares of infaunal species and porcelaneous species are twice as high. Cluster IV comprises all samples from station SO286_75 in the area Seamount with the exemption of sample SO286_75_25 (Figure 4). It is characterized by a high evenness of J = 0.92. The most abundant species *Abditodentrix pseudothalmanni* (Boltovskoy & Guissani de Kahn, 1981) and *Globocassidulina subglobosa* (Brady, 1881) have a relatively low mean share of 11.8% and 10.4% respectively (Table 3). The mean diversity is slightly higher than in cluster III. In comparison to the other clusters the share of infaunal species is significantly higher and that of porcelaneous species is higher (Table 3).

The exceptional samples SO286_42_30 and SO286_75_25 are both from the bottom of their cores and differ significantly from the samples up in the core. In sample SO286_42_30 the dominant species are together more than halved in abundance and partly replaced by *Uvigerina* sp. 2 and *Pullenia quinqueloba*, which are of little significance higher in the core (Supplementary Materials Tables S1 and S2). The evenness of J = 0.88 is much higher than J = 0.73 in the rest of the core (Appendix A Table A4). In sample SO286_75_25 the dominant species *Epistominella exigua has a share of 20%*, whereas above its share ranges from 3% to 9%. (Supplementary Materials Tables S1 and S2). The share of epifaunal species is about 40% higher than in the samples higher in the core.

**Table 3.** Main characteristics of the four clusters shown in Figure 4. The underlying data for diversity and evenness are given in Appendix A Table A4, for species names and importance, life position and wall material in Supplementary Materials Tables S1 and S2 and for density in Appendix A Table A2.

| Cluster | | I | | II | | III | | IV | |
|---|---|---|---|---|---|---|---|---|---|
| Corresponding with station SO286_ | | 21 | | 42 | | 65,66,67 | | 75 | |
| Area | | Labrador Sea | | Labrador Basin | | SW of Azores | | Seamount | |
| Dominant species | | *E. exigua* *O. umbonatus* | | *E. exigua* *O. umbonatus* | | *E. exigua* | | None | |
| Subsidiary important species | | *H. elongata* | | *G. communis* | | *C. wuellerstorfi* | | *A. pseudothalmanni* | |
| | | *C. wuellerstorfi* | | | | *Melonis* group | | *G. subglobosa* | |
| | | *P. quinqueloba* | | | | | | | |
| | | Mean | Range | Mean | Range | Mean | Range | Mean | Range |
| Share in % | *E. exigua* | 23.3 | 6.8–32.9 | 39.0 | 30.4–50.6 | 22.7 | 13.3–29.9 | 5.9 | 3.2–9.1 |
| Share in % | *O. umbonatus* | 10.5 | 5.9–19.0 | 13.6 | 8.9–18.4 | 3.2 | 0.0–7.9 | 1.8 | 0.0–2.6 |
| Share in % | *A. pseudothalmanni* | 0.0 | 0.0 | 0.0 | 0.0 | 0.6 | 0.0–1.9 | 11.8 | 2.5–19.5 |
| Share in % | *G. subglobosa* | 0.1 | 0.0–0.6 | 0.0 | 0.0 | 1.9 | 0.0–9.7 | 10.4 | 6.5–15.1 |
| Diversity | Fisher's alpha (α) | 14.6 | 10.5–18.4 | 14.1 | 10.6–17.3 | 23.3 | 16.0–30.1 | 25.9 | 22.7–33.8 |
| Evenness | Equitability (J) | 0.83 | 0.77–0.93 | 0.73 | 0.63–0.79 | 0.87 | 0.81–0.93 | 0.92 | 0.90–0.94 |
| Div. + Even. | Shannon (H) | 2.98 | 2.59–3.44 | 2.60 | 2.28–2.86 | 3.37 | 3.01–3.62 | 3.62 | 3.49–3.72 |
| Life Position | Epifaunal | 65% | 50%–76% | 80% | 70%–83% | 59% | 49%–67% | 37% | 33%–42% |
| Life Position | Infaunal | 15% | 6%–22% | 10% | 7%–17% | 25% | 17%–33% | 39% | 35%–44% |
| Life Position | Unknown/Both | 21% | 13%–32% | 10% | 8%–13% | 16% | 10%–22% | 23% | 16%–29% |
| Wall Material | Organic | 4% | 3%–7% | 1% | 0%–1% | 0% | 0%–1% | 0% | 0%–1% |
| Wall Material | Agglutinated | 27% | 19%–45% | 9% | 5%–17% | 7% | 3%–17% | 5% | 2%–8% |
| Wall Material | Porcelaneous | 4% | 2%–8% | 6% | 4%–8% | 11% | 7%–16% | 16% | 9%–24% |
| Wall Material | Calcareous | 65% | 48%–74% | 85% | 74%–90% | 82% | 68%–88% | 80% | 74%–88% |
| Density per g | | 124 | 44–290 | 889 | 438–1059 | 279 | 163–447 | 189 | 110–223 |

### 3.3. Trends in the Assemblages down the Cores

Only the samples from station SO286_21 show a clear trend down the core. The diversity declines from $\alpha = 17.05$ in the top layer to $\alpha = 10.52$ in the lowest part, as well as the evenness from $J = 0.93$ to $J = 0.77$ (Appendix A Table A4). The share of epifaunal taxa rises down the core from 50.3% to 76.1%, and that of calcareous species from 48.4% to 73.5% (Supplementary Materials Table S1). Such trends are not observable in the other cores. The exceptional samples SO286_42_30 and SO286_75_25 show, as already discussed, an abrupt change in their species composition in the deepest slice.

## 4. Discussion

### 4.1. Limitations and Advantages of This Study

The results of this study are biased by taphonomic effects. It is known that organic walled specimens rapidly decay and thin walled agglutinated forms easily disintegrate [38]. This caused the lack of organic walled taxa in the investigated material besides of a few *Placopsilinella aurantiaca* Earland, 1934, which were well preserved. It will also have caused an underestimation of agglutinated taxa. The key species *E. exigua* may not be satisfactorily sampled if the >125 μm fraction is studied, as there might be smaller adult specimens [39].

As the number of specimens alive was very low (Appendix A Table A2) this study is based on total assemblages (dead and alive faunas). The composition of a live fauna reflects the conditions at the time of collection, which is influenced by seasonality and other momentary biases, whereas here material is time averaged. For abyssal plains the rate of sedimentation is known to be very low as the sediment is almost exclusively of organic origin. For example for the North East Atlantic rates between 0.14 cm and 3.2 cm

per 1000 years are reported [40]. The investigated slices of 2 cm may represent up to about 14,000 years and those of 5 cm up to about 35,000 years. The studied faunas are interpreted to represent the overall faunal composition in the Holocene and late Pleistocene.

Biological productivity in abyssal plains is reported to be very low, but the biological activity can still create a bioturbated sediment mixed layer of 5–11 cm [40]. Lower parts of the core may also be affected by bioturbation in the timespan when they built the top sediment. Trends in the assemblages down the core will not only reflect faunal changes over time but are also influenced by bioturbation and taphonomic loss. In this study it is concluded that only the samples from station SO286_21 show a clear trend towards lower diversity and higher shares of epifaunal and calcareous taxa down the core. Furthermore the abrupt changes in the faunal composition of the deepest slices in core SO286_42 and SO286_75 are seen as indicative of a change in environmental conditions. In contrast to the limited results for the down core investigations the faunal compositions between the working areas are shown to differ substantially. Four explicitly separate clusters were found (Figure 4) with specific characteristics (Table 3).

The density of benthic foraminifera was measured for each sample by determining their amount in 1 g of dry sediment (Dst in Appendix A Table A2). Due to the taphonomic loss and chosen size fraction its usability as a proxy for biological productivity is seen as rather limited though. Most of the living fauna might be neglected. The lowest density was found in the material from the Labrador Sea and the highest in that of the Labrador Basin. An abrupt change was again found for the deepest slices in core SO286_42 and SO286_75 with a significantly lower and respectively higher density than in the rest of the cores. It indicates that the chosen density measures give additional information. Their usability would need an analysis, which goes beyond of this study.

*4.2. Benthic Foraminiferal Associations Dominated by Epistominella exigua*

Clusters I-III are characterized by *E. exigua* dominated associations. A dominance of this epifaunal species is reported from abyssal areas in the Iceland Basin, Irminger Sea, Labrador Sea, Norwegian, and Greenland Seas by [24]. They are reported from abyssal zones worldwide: Norway, NE Atlantic, NW Africa, SW of Azores, Sierra Leone rise, North and Southwest Indian Ocean, Andaman Sea, Pacific Deep Water areas, Nazca Plate, East Pacific Rise, Weddell Sea, Ross Sea, South Sandwich, Amundsen Sea [39], lower parts of the Walvis Ridge, and the Southwest African continental margin, the Angola and Cape Basins, [27] and in general for Southern, Indian, and Atlantic oceans [5]. These faunas have been linked to seasonal detrital carbon flux rather than water masses [41]. They are found at lower bathyal to abyssal but not hadal depths [4,5]. The variations in abundances of *E. exigua* are described to be significant [24] and are correlated to its opportunistic behavior towards detritus flux [4]. Clusters I and II are codominated by *O. umbonatus*, which is also reported to be an opportunistic, epifaunal species of the deep-sea [4]. In Cluster III *E. exigua* is still dominant and is accompanied by the frequent and also epifaunal *C. wuellerstorfi*. The share of infaunal species and diversity *α* are though double as high as in clusters I and II (Table 3), which indicate a different environmental setting.

The cluster analysis of the subsidiary fauna excluding *E. exigua* and *O. umbonatus* reveals a hundred percent correlation of faunal clusters with the four working areas (Figure 5). The clusters are markedly distinct from each other, which indicates that abyssal faunas are quite heterogeneous and not as uniform as expected. The number of dominant species in the studied material is though very limited and followed by a long tail of less abundant species. This observation is consistent with previous studies of abyssal habitats [4,39]. According to Pawlowski et al. [42] *E. exigua*, *O. umbonatus* and *C. wuellerstorfi* show high genetic similarity. A sample transect from the abyssal Labrador Sea through the Labrador Basin towards the Mid-Atlantic Ridge would be required to draw further conclusions.

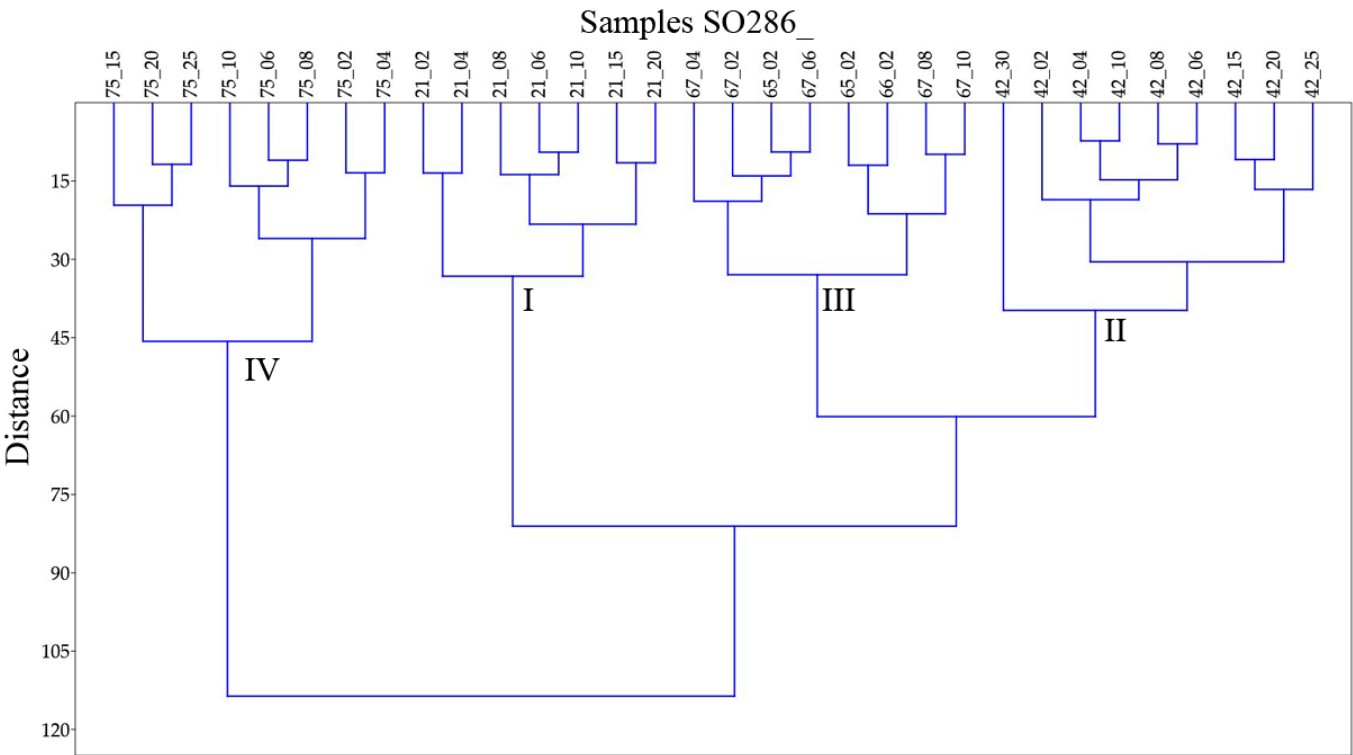

**Figure 5.** Dendrogram of benthic foraminiferal associations in 32 samples from abyssal plains in the Labrador Sea, Labrador Basin and southwest of the Azores collected during RV Sonne cruise SO286 excluding *E. exigua* and *O. umbonatus*. The dendrogram is built using the hierarchical cluster analysis in Q-mode with Ward's method algorithm. It shows the four clusters I, II, III and IV.

*4.3. Benthic Foraminiferal Associations with High Evenness and Lacking Dominant Species*

Cluster IV corresponds with the working area Seamount. It has the greatest distance to the other clusters (Figure 5). The mean diversity α, mean evenness J and share of infaunal species is highest among the clusters (Table 3). There are no dominant species and *E. exigua* and *O. umbonatus* only play a minor role. The most important species *A. pseudothalmanni* and *G. subglobosa* just have a share of 11.8% and 10.4% respectively (Table 3, Supplementary Materials Table S2). The reason for the faunal differences are seen in the different habitat. It is situated at the lower flank of a seamount and more than 690 m above the other working stations. The most abundant species *A. pseudothalmanni* is recorded as a cosmopolitan species from mid-bathyal to upper abyssal depths [5,43]. *G. subglobosa* as the second most abundant species is recorded worldwide from outer shelves, slopes and seamounts (Distribution map in [8]). To draw further conclusions it would be necessary to investigate more than just one point on this seamount flank.

**5. Conclusions**

The foraminiferal faunas of the abyssal North Atlantic are understudied and documented. In this study the faunas at four working areas in the Labrador Sea, Labrador Basin, and southwest of the Azores are for the first time documented and analyzed. One hundred and fifty benthic foraminiferal taxa are identified in counts of 4986 specimens from six cores of ten to thirty cm lengths. One hundred and twenty four taxa are illustrated with optical and/or SEM images on twelve plates.

Four species-based clusters are identified with hierarchical cluster analysis in Q-mode, which correspond to the four working areas. Each cluster is time averaged over periods of ten thousands of years and the observed faunas are not obscured by seasonal or other momentary biases. The clusters show explicitly distinct characteristics, which indicate that abyssal faunas are not uniform but heterogeneous. Three clusters are dominated by

*Epistominella exigua*, which is recorded for many abyssal plains worldwide. The faunal differences are manifested in the long tail of less important species and differing abundances of *E. exigua*. Further studies are needed to get a more detailed picture for the abyssal Northwest Atlantic and understand the driving factors for biodiversity patterns.

**Supplementary Materials:** The following supporting information can be downloaded at https://www.mdpi.com/article/10.3390/d15030381/s1: Table S1: Quantitative abundances of benthic foraminifera in 32 samples of RV Sonne cruise SO286 from six stations in abyssal plains of the Labrador Sea, Labrador Basin and southwest of the Azores. A = agglutinated; C = calcareous; E = epifaunal; I = infaunal; O = organic; P = porcelaneous; U/B = unknown or both; Table S2: Share of benthic foraminifera in 32 samples of RV Sonne cruise SO286 from six stations in abyssal plains of the Labrador Sea, Labrador Basin and southwest of the Azores calculated from Supplementary Table S1. A = agglutinated; C = calcareous; E = epifaunal; I = infaunal; O = organic; P = porcelaneous; U/B = unknown or both.

**Funding:** This research received no further external funding.

**Institutional Review Board Statement:** Not applicable.

**Data Availability Statement:** The counting data will be uploaded to PANGAEA.

**Acknowledgments:** I would like to thank the following people for their key contributions to my manuscript: the crew of RV Sonne during SO286 (IceDivA2) cruise; Alexander Kienecke for retrieving and handling the samples on board; Antje Fischer and Karen Jeskulke (TAs DZMB HH) for their support in sorting and storing the samples; James Taylor in reviewing the manuscript and providing the maps. The author is also very grateful to the anonymous reviewers for their thoughtful and valuable comments that have greatly improved the paper.

**Conflicts of Interest:** The author declares no conflict of interest.

## Appendix A

Figures A1–A12: On the plates each specimen is marked with a letter followed by a scale bar and one to three images showing different views. The scale bar refers to all images, unless indicated differently. The identification of each specimen, its sample station and the size of the scale bar is given in the text below each plate. Explanations on different views of foraminifera are given in general textbooks such as in [36,44].

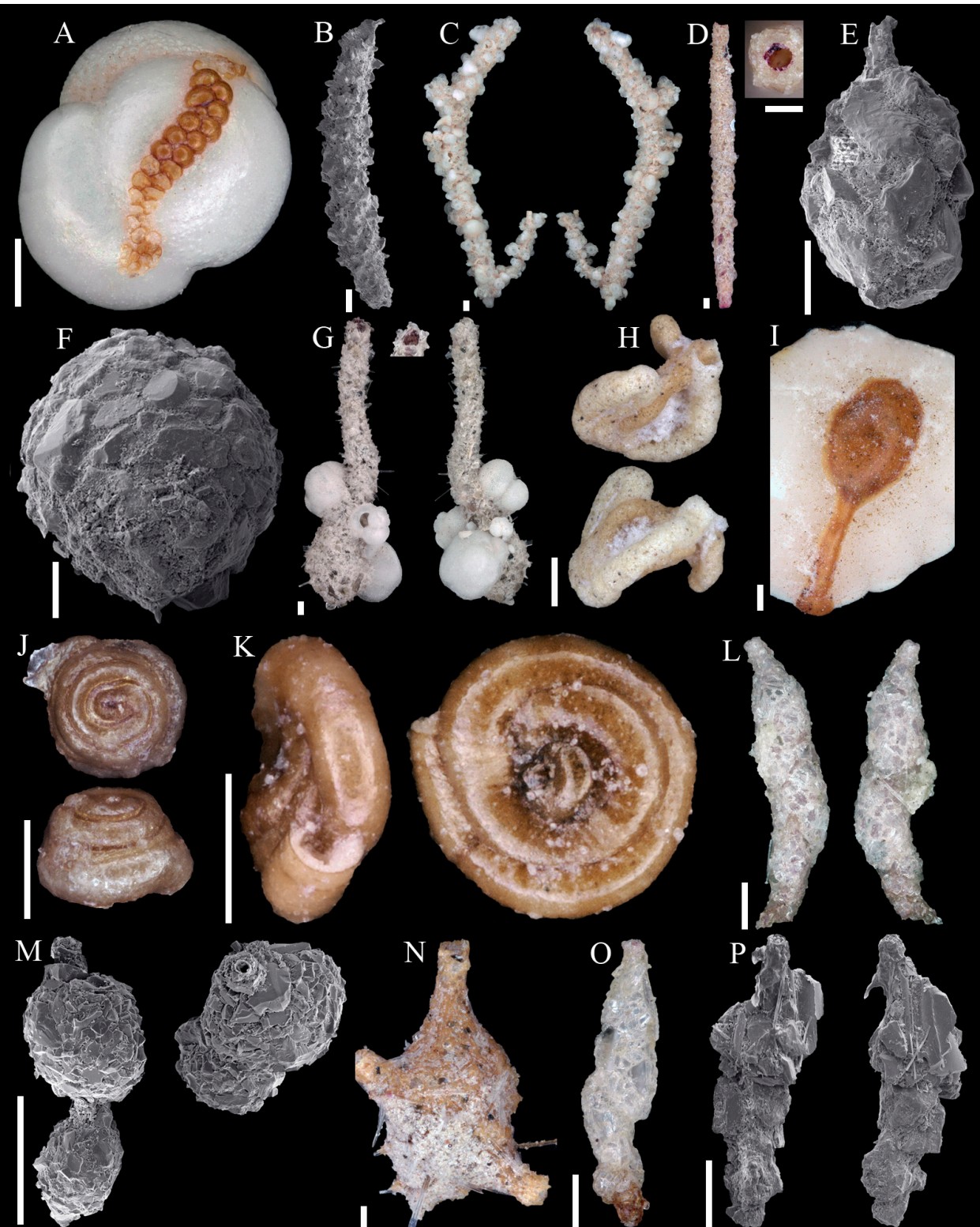

**Figure A1.** Scale bars 100 μm. (**A**) *Placopsilinella aurantiaca* SO286_42; (**B**) Agglutinated tube not assigned to a taxon SO286_42; (**C**) *Rhizammina algaeformis* SO286_42; (**D**) *Hyperammina elongata* SO286_21; (**E**) *Lagenammina arenulata* SO286_42; (**F**) *Psammosphaera fusca* SO286_42; (**G**) *Saccorhiza ramosa* SO286_42; (**H**) *Tolypammina schaudinni* SO286_65; (**I**) *Ammolagena clavata* SO286_65; (**J**) *Glomospira charoides* SO286_75; (**K**) *Glomospira gordialis* SO286_75; (**L,O**) *Reophax* sp. 3 SO286_21; (**M**) *Hormosinelloides guttifer* SO286_42; (**N**) *Aschemonella scabra* SO286_21; (**P**) *Reophax scorpiurus* SO286_21.

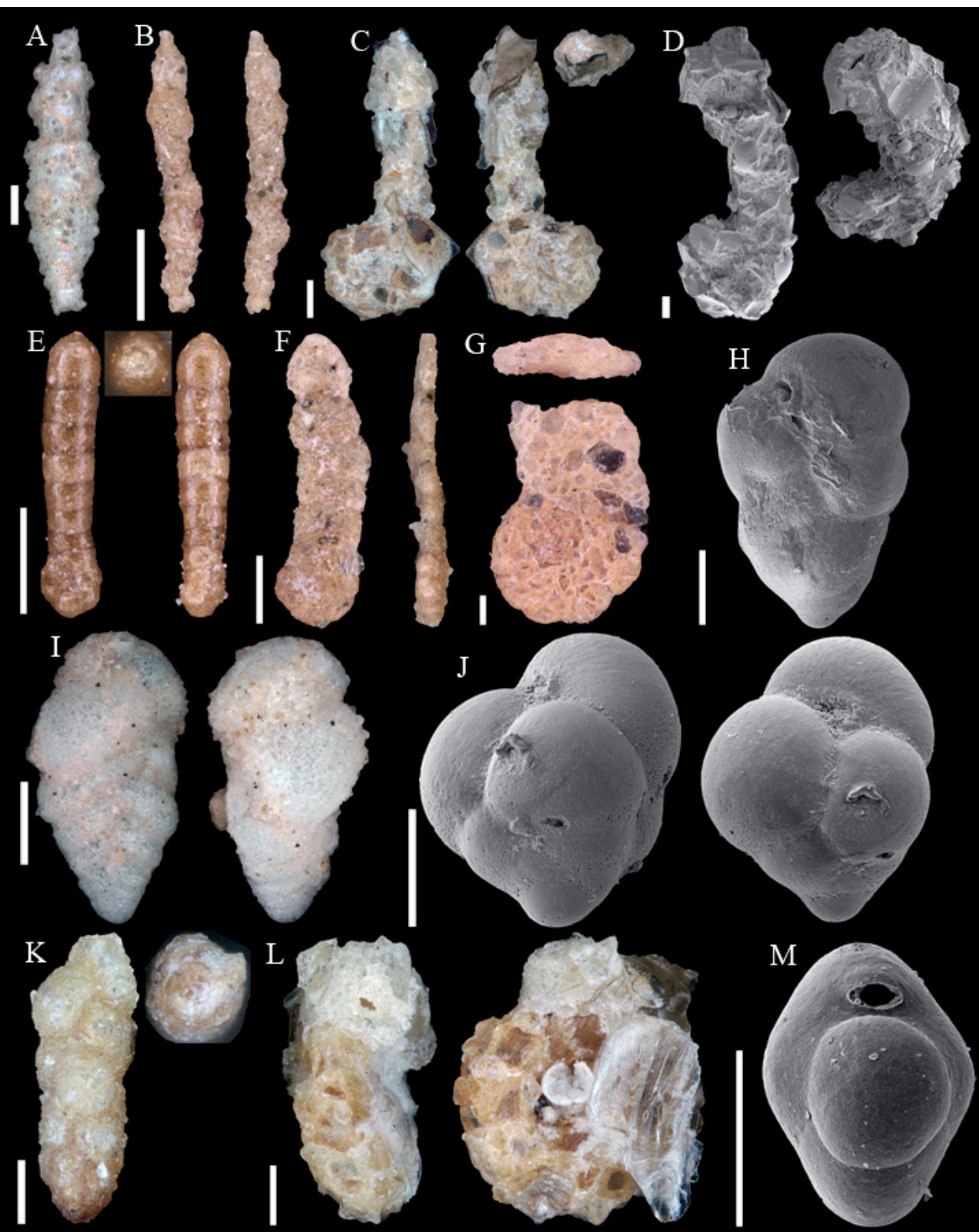

**Figure A2.** Scale bars 100 μm. (**A**) *Reophax* sp. 1 SO286_42; (**B**) *Subreophax aduncus* SO286_67; (**C**) *Ammobaculites agglutinans* SO286_42; (**D**) *Ammobaculites crassaformis* SO286_42; (**E**) *Ammobaculites filiformis* SO286_75; (**F**) *Eratidus foliaceus* SO286_67; (**G**) *Glaphyrammina americana* SO286_42; (**H**) *Karreriella bradyi* SO286_42; (**I**) *Siphotextularia rolshauseni* SO286_42; (**J**) *Eggerella bradyi* SO286_42; (**K**) *Karrerulina conversa* SO286_42; (**L**) *Ammobaculites ?* sp. SO286_42; (**M**) *Buzasina galeata* SO286_42.

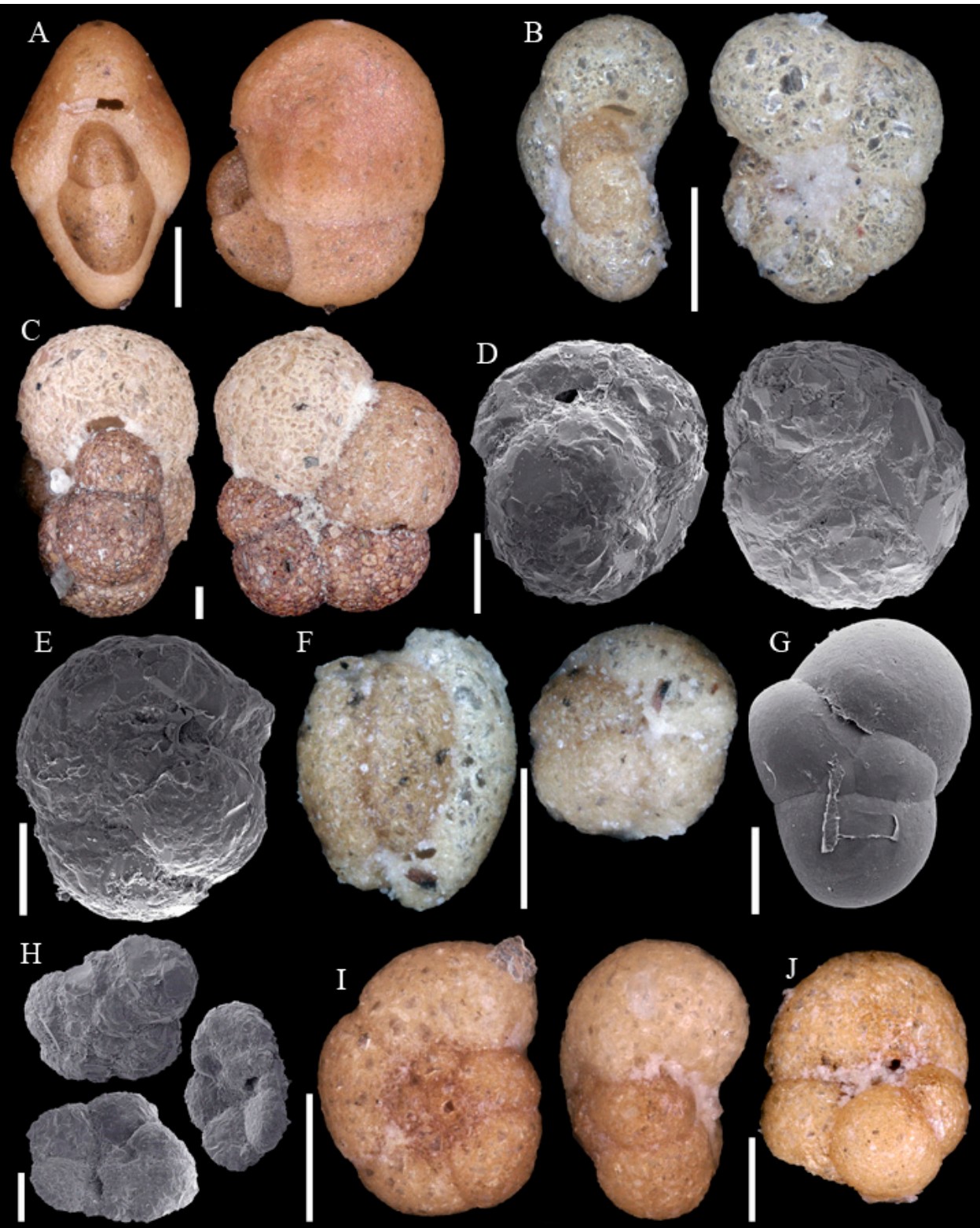

**Figure A3.** Scale bars 100 μm. (**A**) Buzasina ringens SO286_42; (**B**) Cribrostomoides jeffreysii SO286_42; (**C**) Cribrostomoides sphaerilocula SO286_42; (**D**) Cribrostomoides subglobosus SO286_42; (**E**) Recurvoides contortus SO286_42; (**F**) Adercotryma glomeratum SO286_67; (**G**) Cystammina pauciloculata SO286_42; (**H**) Tritaxis heronalleni SO286_75; (**I**) Portatrochammina sp. SO286_75; (**J**) Ammoglobigerina globulosa SO286_42.

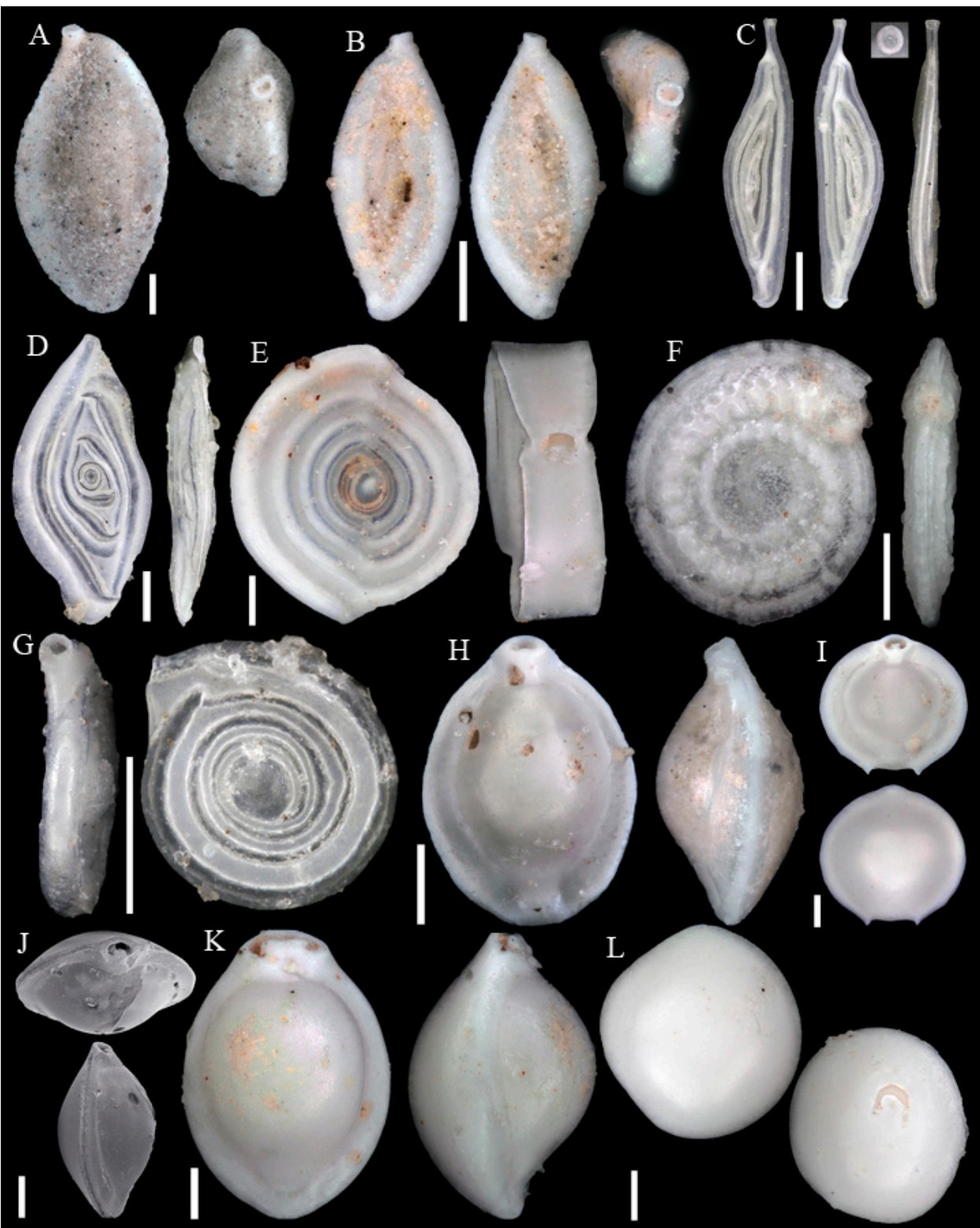

**Figure A4.** Scale bars 100 μm. (**A**) *Sigmoilopsis schlumbergeri* SO286_42; (**B**) *Ammomassilina alveoliniformis* SO286_42; (**C**) *Spirosigmoilina pusilla* SO286_67; (**D**) *Spirophthalmidium acutimargo* SO286_75; (**E**) *Spiroloculina excavata* SO286_75; (**F**) *Cornuspira carinata* SO286_75; (**G**) *Cornuloculina inconstans* SO286_75; (**H**): *Pyrgo lucernula* SO286_42; (**I**) *Pyrgo murrhina* SO286_42; (**J**) *Pyrgo simplex* SO286_42; (**K**) *Pyrgo williamsoni* SO286_42; (**L**) *Pyrgoella* sp. SO286_42.

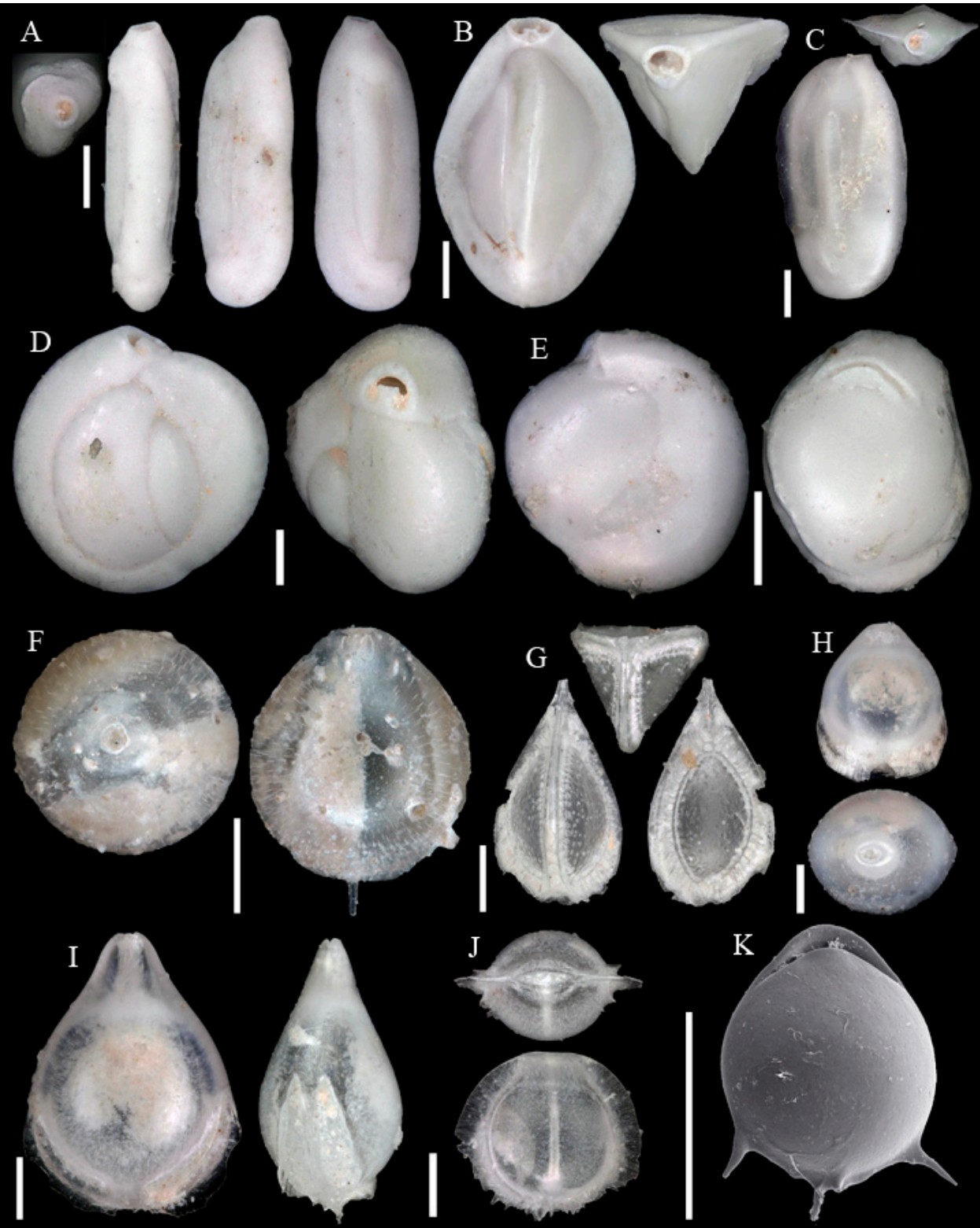

**Figure A5.** Scale bars 100 μm. (**A**) Triloculina oblonga SO286_75; (**B**) Triloculina trihedra SO286_42; (**C**) Quinqueloculina venusta SO286_75; (**D**) Quinqueloculina vulgaris SO286_75; (**E**) Miliolinella subrotunda SO286_75; (**F**) Oolina globosa SO286_42; (**G**) Galwayella trigonoornata SO286_75; (**H**) Fissurina castanea SO286_42; (**I**) Fissurina granifera trimarginata SO286_42; (**J**) Fissurina orbignyana var. rhumbleri SO286_42; (**K**) Fissurina staphyllearia SO286_42.

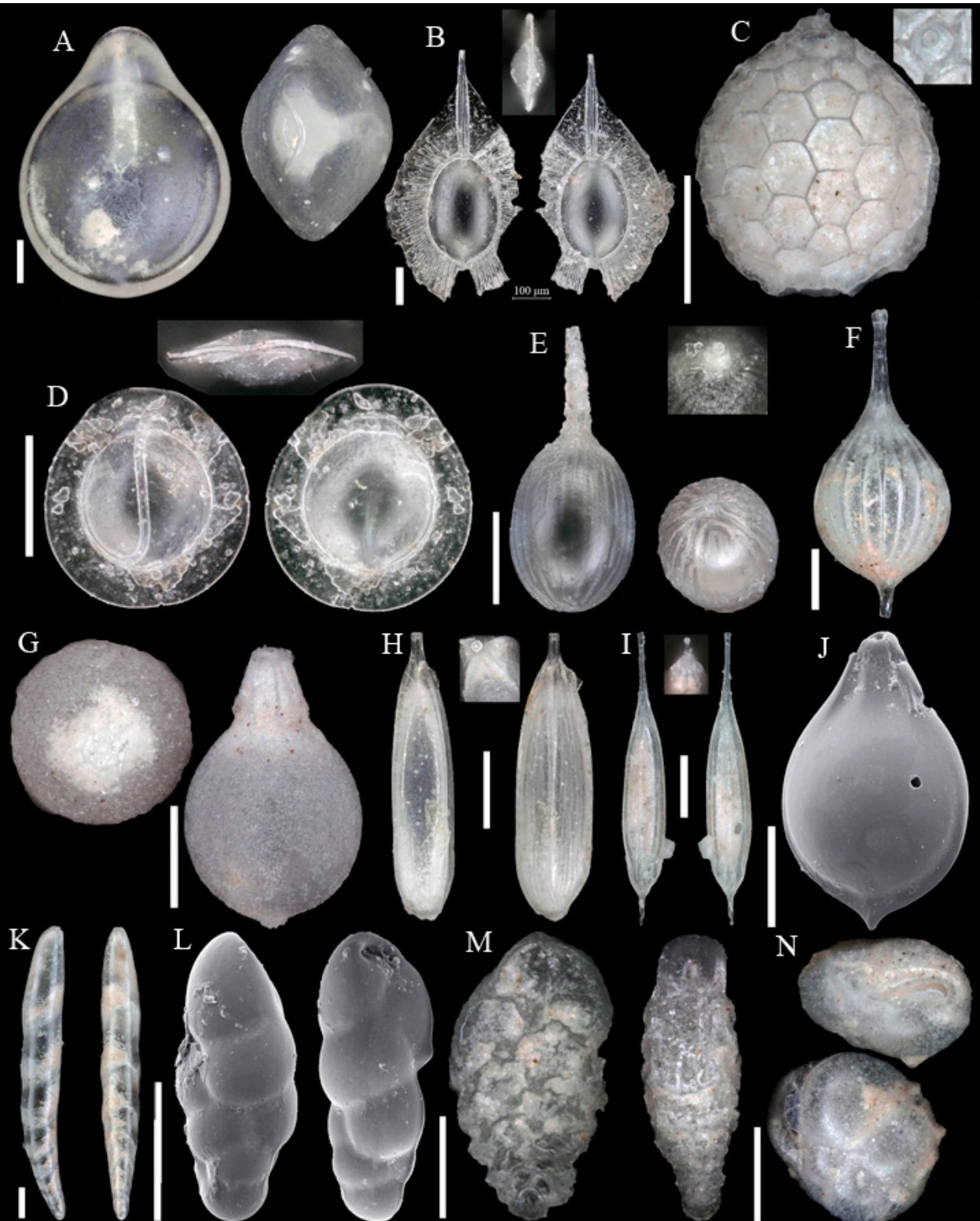

**Figure A6.** Scale bars 100 μm. (**A**) *Fissurina* sp. 1 SO286_67; (**B**) *Lagnea radiata* SO286_67; (**C**) *Favulina hexagona* SO286_75; (**D**) *Lagena wiesneri* SO286_75; (**E**): *Lagena striata* SO286_75; (**F**) *Lagena sulcata* SO286_42; (**G**) *Lagena* sp. 1 SO286_65; (**H**) *Lagena* sp. 2 SO286_42; (**I**) *Procerolagena gracilis* SO286_75; (**J**) *Lagenosolenia incomposita* SO286_42; (**K**) *Laevidentalina haueri* SO286_42; (**L**) *Fursenkoina texturata* SO286_42; (**M**) *Abditodentrix pseudothalmanni* SO286_75; (**N**) *Cassidulina reniforme* SO286_42.

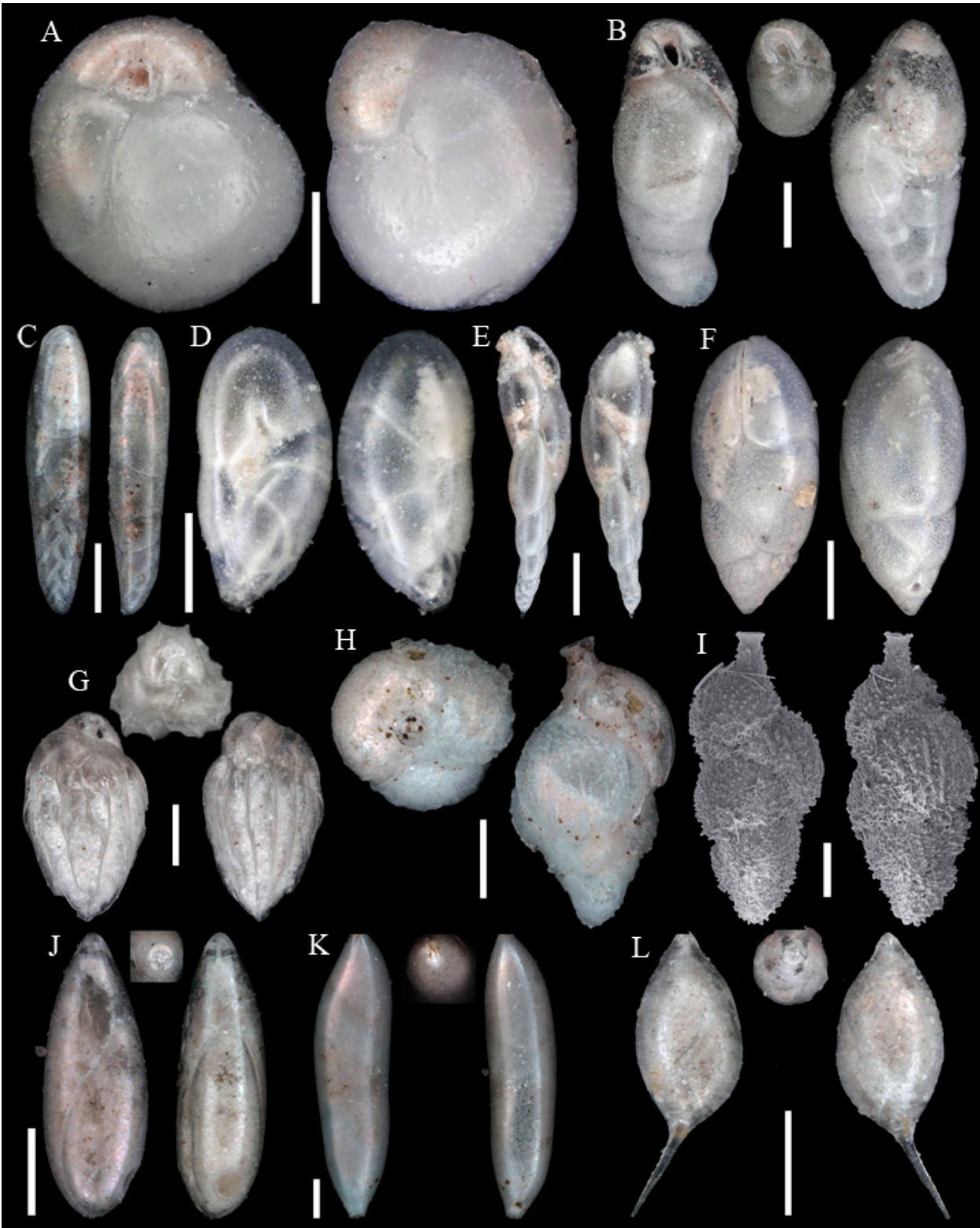

**Figure A7.** Scale bars 100 μm. (**A**) *Globocassidulina subglobosa* SO286_42; (**B**) *Rutherfordoides rotundiformis* SO286_75; (**C**) *Rutherfordoides rotundatus* SO286_67; (**D**) *Robertinoides bradyi* SO286_75; (**E**) *Eubuliminella exilis* SO286_42; (**F**) *Protoglobobulimina* sp. SO286_42; (**G**) *Bulimina buchiana* SO286_75; (**H**): *Uvigerina* sp. 1 SO286_65; (**I**) *Uvigerina* sp. 2 SO286_42; (**J**) *Pyrulina angusta* SO286_75; (**K**) *Pyrulina cylindroides* SO286_75; (**L**) *Pyrulina fusiformis* SO286_75.

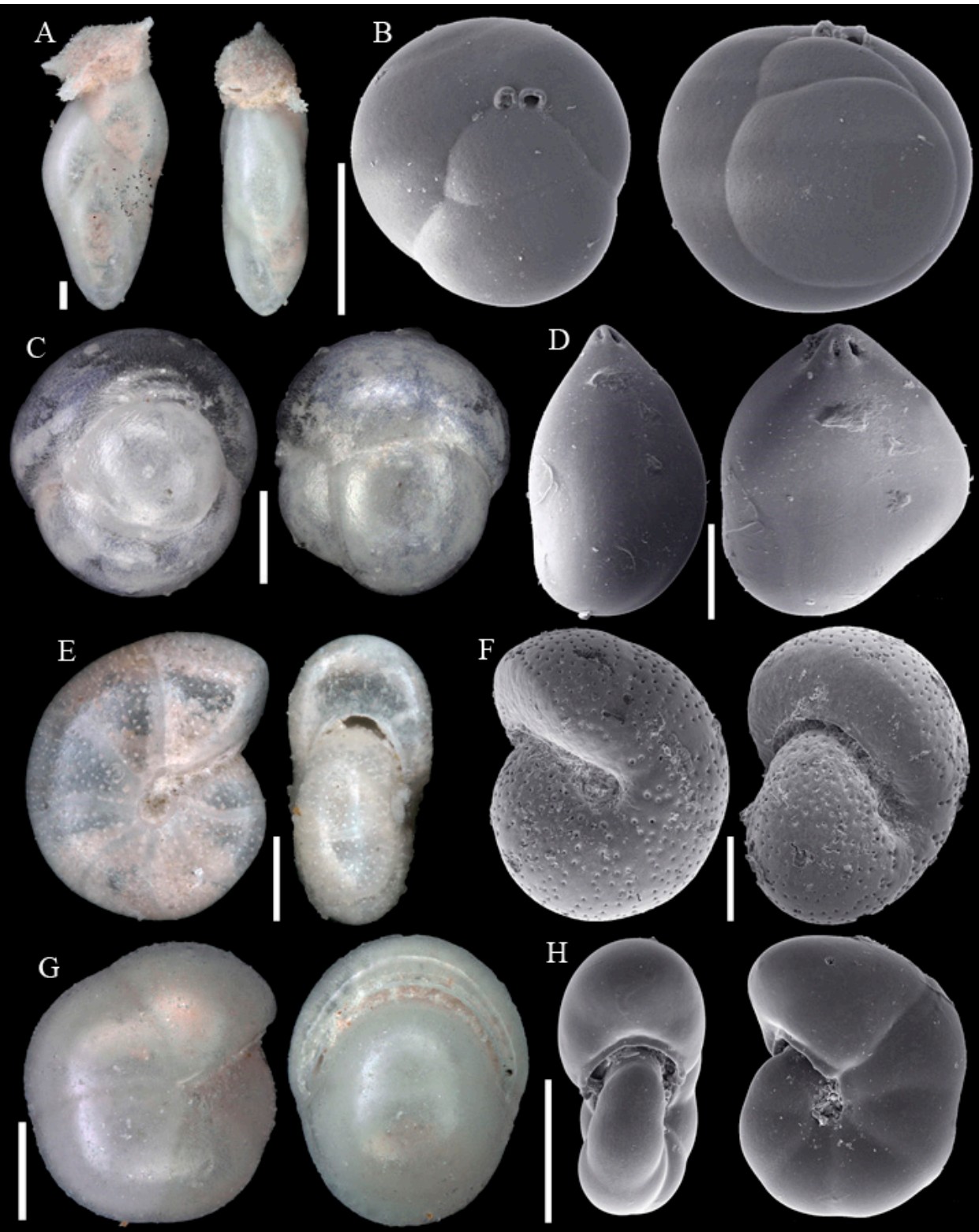

**Figure A8.** Scale bars 100 μm. (**A**) *Pseudopolymorphina novangliae* SO286_42; (**B**) *Eusphaeroidina inflata* SO286_42; (**C**) *Sphaeroidina bulloides* SO286_65; (**D**) *Guttulina communis* SO286_42; (**E**) *Melonis affinis* SO286_42; (**F**) *Melonis pompilioides* SO286_42; (**G**) *Pullenia bulloides* SO286_67; (**H**) *Pullenia quinqueloba* SO286_67.

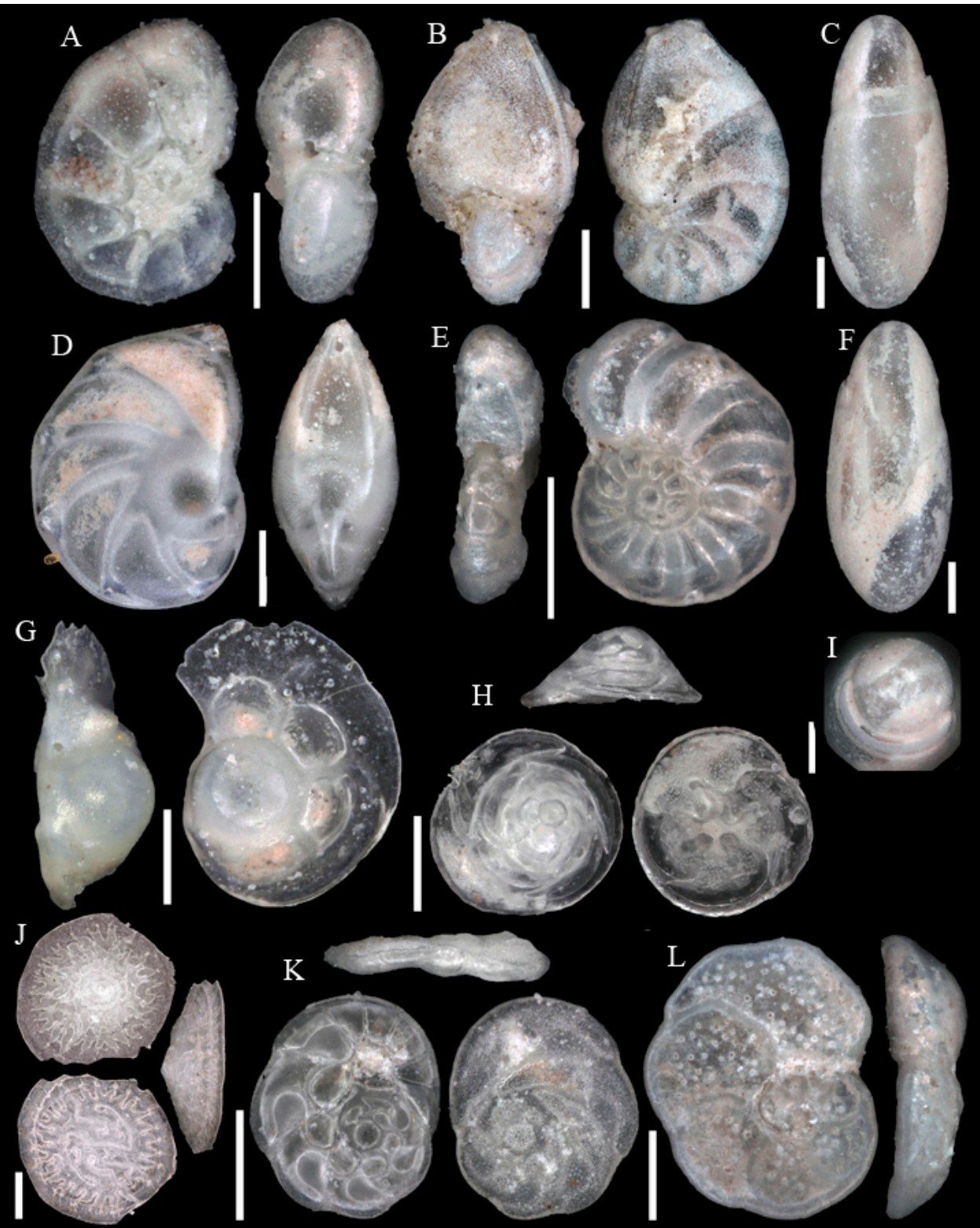

**Figure A9.** Scale bars 100 μm. (**A**) *Pseudononion granuloumbilicatum* SO286_75; (**B**) *Nonionellina labradorica* SO286_42; (**C,F,I**) *Chilostomella oolina* SO286_67; (**D**) *Lenticulina convergens* SO286_67; (**E**) *Hyalinea balthica* SO286_42; (**G**) *Laticarinina pauperata* SO286_42; (**H**) *Patellina simplissima* SO286_75; (**J**) *Patellina corrugata* SO286_42; (**K**) *Discorbinella complanata* SO286_67; (**L**) *Lobatula lobatula* SO286_42.

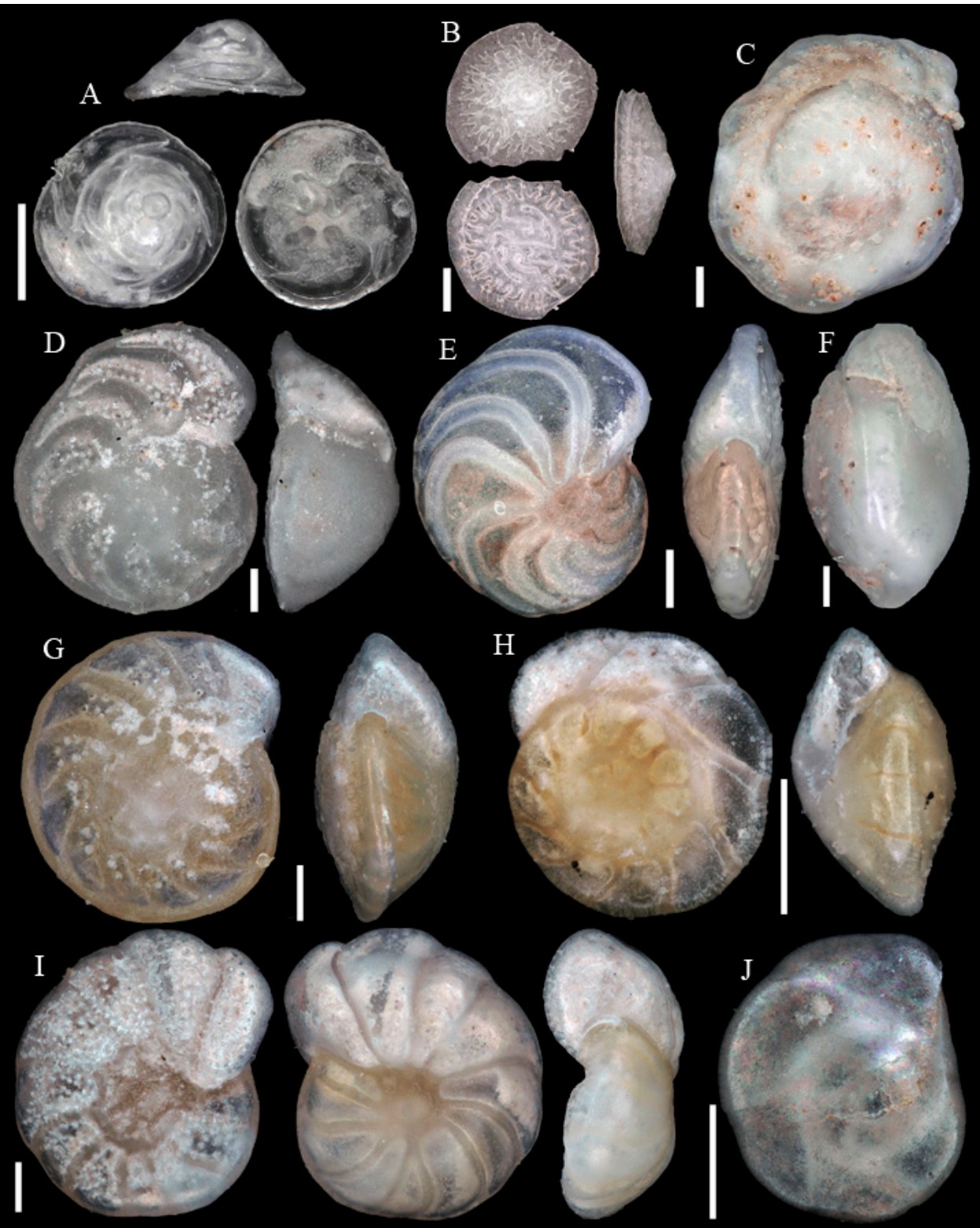

**Figure A10.** Scale bars 100 μm. (**A**) *Patellina simplissima* SO286_75; (**B**) *Patellina corrugata* SO286_42; (**C,F**) *Cibicides pachyderma* SO286_42; (**D**) *Cibicides refulgens* SO286_75; (**E**) *Cibicidoides wuellerstorfi* SO286_42; (**G**) *Cibicidoides mundulus* SO286_75; (**H**) *Gyroidina* sp 1 SO286_75; (**I**) *Cibicidoides cicatricosus* SO286_65; (**J**) *Eponides repandus* SO286_42.

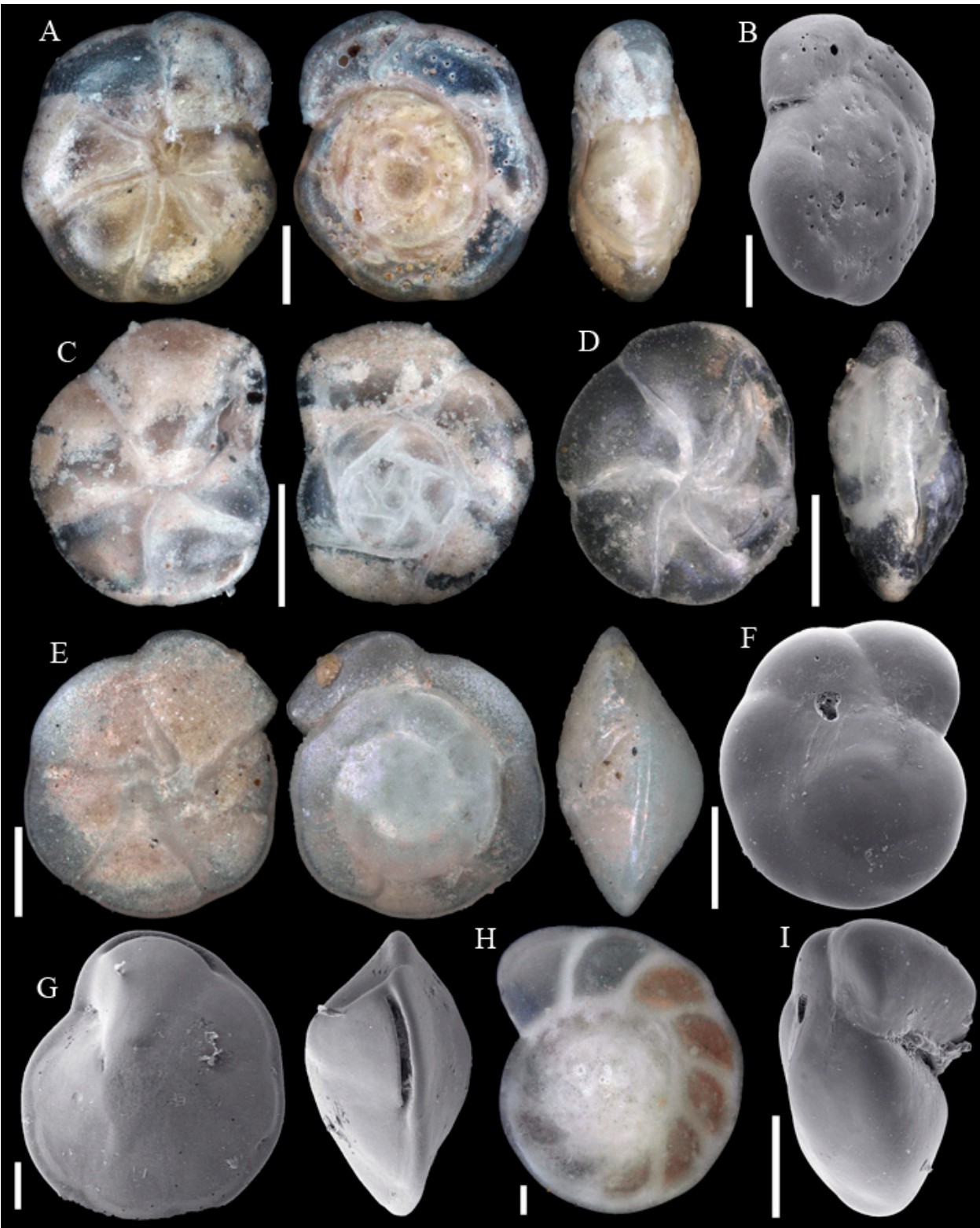

**Figure A11.** Scale bars 100 µm. (**A**) *Alabaminella weddellensis* SO286_42; (**B**) *Alabaminella weddellensis* SO286_42; (**C**) *Epistominella exigua* SO286_42; (**D**) *Epistominella exigua* SO286_42; (**E**) *Oridorsalis umbonatus* SO286_42; (**F,I**) *Gyroidina* sp. 2 SO286_42; (**G**) *Hoeglundina elegans* SO286_42; (**H**) *Hoeglundina elegans* SO286_42.

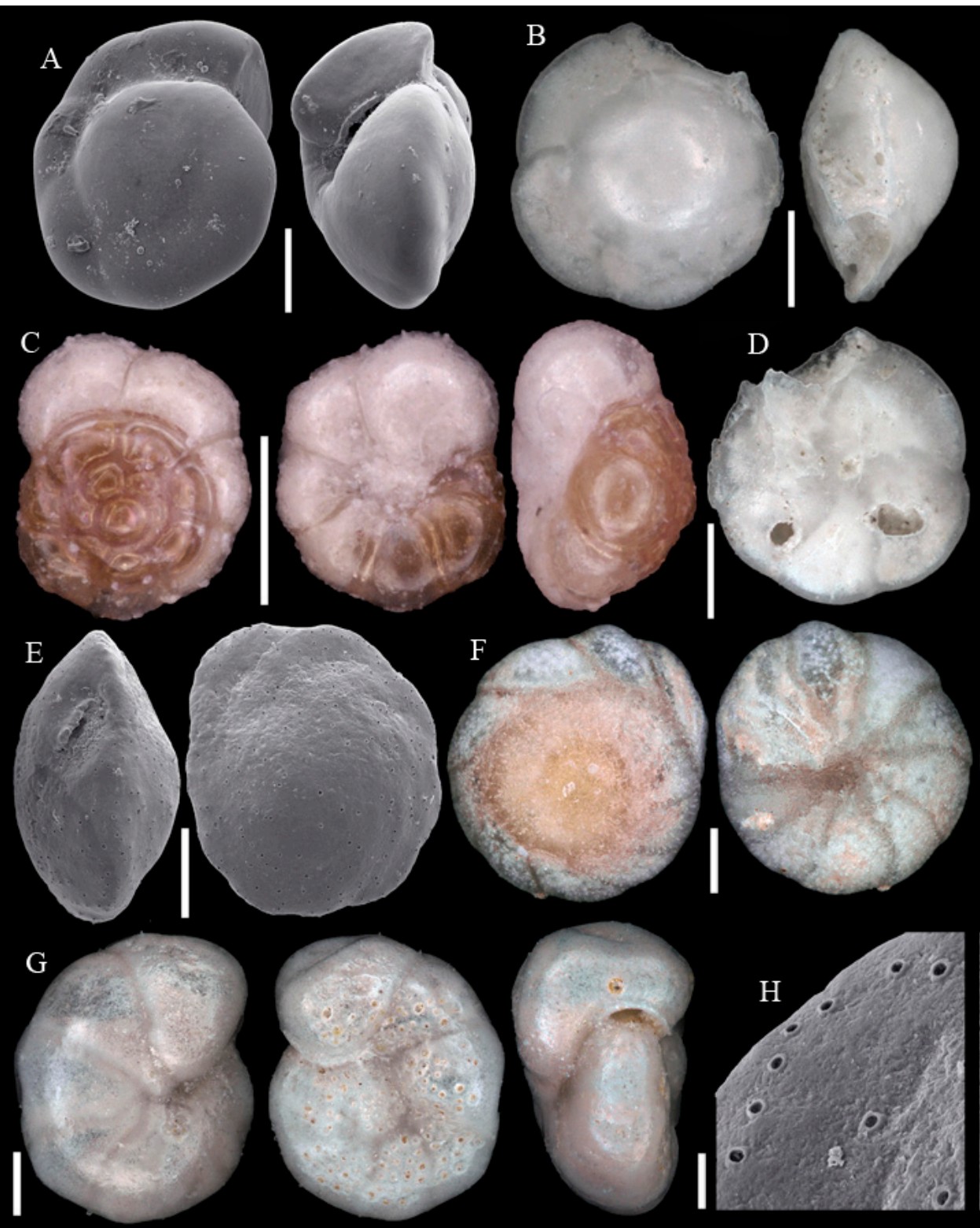

**Figure A12.** Scale bars 100 µm, except for (**H**) 10 µm. (**A**) *Hansenisca soldanii* SO286_42; (**B**,**D**) *Gavelinopsis* sp. SO286_42; (**C**) *Ioanella tumidula* SO286_75; (**E**) *Nuttallides decorata* SO286_42; (**F**) *Nuttallides decorata* SO286_42; (**G**) *Anomalinoides globulosus* SO286_67; (**H**) *Nuttallides decorata* SO286_42.

**Table A1.** Details for the investigated samples of RV Sonne cruise SO286. Slice in cm. ASW = Southwest of the Azores; C = Core; Dev = Device; Go = Globigerina ooze; MUC = Multicorer. Abbreviations: Lat = Latitude; Lon = Longitude; LB = Labrador Basin; LS = Labrador Sea; Sed = Sediment type; SM = Seamount; smu = soft mud; Sta = Station SO286_; WA = Working Area.

| Sample | Sta | C | Slice | WA | Lon | Lat | Depth | Dev | Sed | Facies |
|--------|-----|---|-------|-----|-----|-----|-------|-----|-----|--------|
| SO286_21_02 | 21 | 5 | 0–2 | LS | 58°14.422′ | 54°13.075′ | 3391 m | MUC | smu | Go |
| SO286_21_04 | 21 | 5 | 2–4 | LS | 58°14.422′ | 54°13.075′ | 3391 m | MUC | smu | Go |
| SO286_21_06 | 21 | 5 | 4–6 | LS | 58°14.422′ | 54°13.075′ | 3391 m | MUC | smu | Go |
| SO286_21_08 | 21 | 5 | 6–8 | LS | 58°14.422′ | 54°13.075′ | 3391 m | MUC | smu | Go |
| SO286_21_10 | 21 | 5 | 8–10 | LS | 58°14.422′ | 54°13.075′ | 3391 m | MUC | smu | Go |
| SO286_21_15 | 21 | 5 | 10–15 | LS | 58°14.422′ | 54°13.075′ | 3391 m | MUC | smu | Go |
| SO286_21_20 | 21 | 5 | 15–20 | LS | 58°14.422′ | 54°13.075′ | 3391 m | MUC | smu | Go |
| SO286_42_02 | 42 | 1 | 0–2 | LB | 51°58.255′ | 38°59.534′ | 3685 m | MUC | smu | Go |
| SO286_42_04 | 42 | 1 | 2–4 | LB | 51°58.255′ | 38°59.534′ | 3685 m | MUC | smu | Go |
| SO286_42_06 | 42 | 1 | 4–6 | LB | 51°58.255′ | 38°59.534′ | 3685 m | MUC | smu | Go |
| SO286_42_08 | 42 | 1 | 6–8 | LB | 51°58.255′ | 38°59.534′ | 3685 m | MUC | smu | Go |
| SO286_42_10 | 42 | 1 | 8–10 | LB | 51°58.255′ | 38°59.534′ | 3685 m | MUC | smu | Go |
| SO286_42_15 | 42 | 1 | 10–15 | LB | 51°58.255′ | 38°59.534′ | 3685 m | MUC | smu | Go |
| SO286_42_20 | 42 | 1 | 15–20 | LB | 51°58.255′ | 38°59.534′ | 3685 m | MUC | smu | Go |
| SO286_42_25 | 42 | 1 | 20–25 | LB | 51°58.255′ | 38°59.534′ | 3685 m | MUC | smu | Go |
| SO286_42_30 | 42 | 1 | 25–30 | LB | 51°58.255′ | 38°59.534′ | 3685 m | MUC | smu | Go |
| SO286_65_02 | 65 | 9 | full | ASW | 37°00.025′ | 35°29.491′ | 3193 m | MUC | smu | Go |
| SO286_65_02 | 65 | 11 | full | ASW | 37°00.025′ | 35°29.491′ | 3193 m | MUC | smu | Go |
| SO286_66_02 | 66 | 18 | full | ASW | 37°00.032′ | 35°29.441′ | 3192 m | MUC | smu | Go |
| SO286_67_02 | 67 | 11 | 0–2 | ASW | 37°00.037′ | 35°29.382′ | 3209 m | MUC | smu | Go |
| SO286_67_04 | 67 | 11 | 2–4 | ASW | 37°00.037′ | 35°29.382′ | 3209 m | MUC | smu | Go |
| SO286_67_06 | 67 | 11 | 4–6 | ASW | 37°00.037′ | 35°29.382′ | 3209 m | MUC | smu | Go |
| SO286_67_08 | 67 | 11 | 6–8 | ASW | 37°00.037′ | 35°29.382′ | 3209 m | MUC | smu | Go |
| SO286_67_10 | 67 | 11 | 8–10 | ASW | 37°00.037′ | 35°29.382′ | 3209 m | MUC | smu | Go |
| SO286_75_02 | 75 | 11 | 0–2 | SM | 37°13.922′ | 35°32.316′ | 2508 m | MUC | smu | Go |
| SO286_75_04 | 75 | 11 | 2–4 | SM | 37°13.922′ | 35°32.316′ | 2508 m | MUC | smu | Go |
| SO286_75_06 | 75 | 11 | 4–6 | SM | 37°13.922′ | 35°32.316′ | 2508 m | MUC | smu | Go |
| SO286_75_08 | 75 | 11 | 6–8 | SM | 37°13.922′ | 35°32.316′ | 2508 m | MUC | smu | Go |
| SO286_75_10 | 75 | 11 | 8–10 | SM | 37°13.922′ | 35°32.316′ | 2508 m | MUC | smu | Go |
| SO286_75_15 | 75 | 11 | 10–15 | SM | 37°13.922′ | 35°32.316′ | 2508 m | MUC | smu | Go |
| SO286_75_20 | 75 | 11 | 15–20 | SM | 37°13.922′ | 35°32.316′ | 2508 m | MUC | smu | Go |
| SO286_75_25 | 75 | 11 | 20–25 | SM | 37°13.922′ | 35°32.316′ | 2508 m | MUC | smu | Go |

**Table A2.** Weights, amounts and density of specimens for samples of RV Sonne cruise SO286. Slice in cm from top of the core. Abbreviations: a-4 = at −4° C; Asa = Amount of specimens alive; Ast = Amount of specimens dead and alive; c = calculated; Dsa = Density of specimens alive per 100 g; Dst = Density of specimens alive and dead per 100 g; m = measured; Pa = Part analysed; Pr = Part retained; wf = washed fraction >125 μm and <2000 μm; Ssa = Subsample alive; Sst = Subsampled dead and alive; Wg = Weight in gram.

| Measure | | Wg | Wg | Wg | Wg | Wg | Wg | Wg | Asa | Dsa | Wg | Ast | Dst |
|---------|---|----|----|----|----|----|----|----|-----|-----|----|-----|-----|
| Part | | whole | Pa | Pr | Pa | Pr | Pa | Ssa | Ssa | Ssa | Sst | Sst | Sst |
| Phase | | a-4 | a-4 | a-4 | dry | dry | dry | dry | dry | dry | dry | dry | dry |
| Fraction | | full | full | full | full | full | wf | wf | wf | full | wf | wf | full |
| Sample | Slice | c | m | m | c | m | m | m | m | c | m | m | c |

**Table A2.** *Cont.*

| Measure | | Wg | Wg | Wg | Wg | Wg | Wg | Wg | Asa | Dsa | Wg | Ast | Dst |
|---|---|---|---|---|---|---|---|---|---|---|---|---|---|
| Column | | 1 | 2 | 3 | 4 | 5 | 6 | 7 | 8 | 9 | 10 | 11 | 12 |
| Calculation | | 2 + 3 | | | 5/3*2 | | | | | 8/7*6/4 | | | 11/10*6/4 |
| SO286_21_02 | 0–2 | 157.2 | 119.1 | 38.1 | 41.0 | 13.1 | 3.7 | 0.1 | 5 | 5 | 0.1 | 161 | 145 |
| SO286_21_04 | 2–4 | 153.3 | 134.9 | 18.4 | 41.1 | 5.6 | 3.5 | 0.1 | 2 | 2 | 0.1 | 153 | 130 |
| SO286_21_06 | 4–6 | 116.7 | 92.5 | 24.2 | 39.8 | 10.4 | 7.6 | 0.1 | 1 | 2 | 0.1 | 152 | 290 |
| SO286_21_08 | 6–8 | 203.9 | 167.7 | 36.2 | 90.8 | 19.6 | 4.7 | 0.1 | 0 | 0 | 0.1 | 161 | 83 |
| SO286_21_10 | 8–10 | 158.1 | 137.1 | 21.0 | 78.3 | 12.0 | 4.5 | 0.1 | 0 | 0 | 0.1 | 153 | 88 |
| SO286_21_15 | 10–15 | 372.6 | 334.3 | 38.3 | 178.9 | 20.5 | 9.6 | 0.1 | 0 | 0 | 0.1 | 163 | 87 |
| SO286_21_20 | 15–20 | 301.1 | 227.0 | 74.1 | 123.5 | 40.3 | 7.0 | 0.2 | 0 | 0 | 0.2 | 155 | 44 |
| SO286_42_02 | 0–2 | 126.0 | 85.8 | 40.2 | 43.5 | 20.4 | 24.1 | 0.2 | 1 | 3 | 0.2 | 158 | 438 |
| SO286_42_04 | 2–4 | 179.4 | 130.2 | 49.2 | 55.0 | 20.8 | 36.4 | 0.1 | 0 | 0 | 0.1 | 160 | 1059 |
| SO286_42_06 | 4–6 | 193.1 | 153.6 | 39.5 | 75.8 | 19.5 | 43.1 | 0.1 | 2 | 11 | 0.1 | 157 | 893 |
| SO286_42_08 | 6–8 | 174.1 | 147.5 | 26.6 | 70.4 | 12.7 | 43.6 | 0.1 | 0 | 0 | 0.1 | 153 | 948 |
| SO286_42_10 | 8–10 | 210.2 | 161.5 | 48.7 | 85.6 | 25.8 | 50.1 | 0.1 | 0 | 0 | 0.1 | 155 | 907 |
| SO286_42_15 | 10–15 | 326.8 | 278.5 | 48.3 | 148.2 | 25.7 | 84.7 | 0.1 | 0 | 0 | 0.1 | 154 | 880 |
| SO286_42_20 | 15–20 | 288.3 | 232.7 | 55.6 | 138.5 | 33.1 | 85.3 | 0.1 | 0 | 0 | 0.1 | 158 | 973 |
| SO286_42_25 | 20–25 | 418.1 | 349.2 | 68.9 | 198.2 | 39.1 | 63.4 | 0.1 | 0 | 0 | 0.1 | 159 | 1017 |
| SO286_42_30 | 25–30 | 191.9 | 156.5 | 35.4 | 81.8 | 18.5 | 47.6 | 0.2 | 0 | 0 | 0.2 | 152 | 442 |
| SO286_65_09 | full core | 178.2 | 160.9 | 17.3 | 79.1 | 8.5 | 49.0 | 0.3 | 1 | 2 | 0.3 | 152 | 314 |
| SO286_65_11 | full core | 220.8 | 185.7 | 35.1 | 98.9 | 18.7 | 56.3 | 0.4 | 4 | 6 | 0.4 | 154 | 219 |
| SO286_66 | full core | 602.1 | 568.5 | 33.6 | 311.3 | 18.4 | 167.4 | 0.5 | 0 | 0 | 0.5 | 156 | 168 |
| SO286_67_02 | 0–2 | 209.8 | 138.8 | 71.0 | 73.7 | 37.7 | 39.2 | 0.4 | 4 | 5 | 0.4 | 159 | 211 |
| SO286_67_04 | 2–4 | 258.9 | 215.3 | 43.6 | 114.6 | 23.2 | 66.4 | 0.3 | 0 | 0 | 0.3 | 157 | 303 |
| SO286_67_06 | 4–6 | 274.8 | 221.2 | 53.6 | 128.3 | 31.1 | 69.6 | 0.3 | 0 | 0 | 0.3 | 154 | 278 |
| SO286_67_08 | 6–8 | 180.9 | 138.3 | 42.6 | 68.8 | 21.2 | 45.2 | 0.4 | 0 | 0 | 0.4 | 150 | 246 |
| SO286_67_10 | 8–10 | 228.1 | 187.8 | 40.3 | 101.6 | 21.8 | 63.4 | 0.6 | 0 | 0 | 0.6 | 157 | 163 |
| SO286_75_02 | 0–2 | 207.1 | 167.0 | 40.1 | 77.9 | 18.7 | 28.9 | 0.3 | 4 | 5 | 0.3 | 154 | 190 |
| SO286_75_04 | 2–4 | 182.4 | 145.1 | 37.3 | 78.2 | 20.1 | 26.7 | 0.3 | 0 | 0 | 0.3 | 154 | 175 |
| SO286_75_06 | 4–6 | 122.5 | 80.2 | 42.3 | 44.6 | 23.5 | 14.3 | 0.3 | 0 | 0 | 0.3 | 151 | 161 |
| SO286_75_08 | 6–8 | 172.4 | 134.1 | 38.3 | 70.4 | 20.1 | 23.5 | 0.4 | 0 | 0 | 0.4 | 159 | 133 |
| SO286_75_10 | 8–10 | 236.1 | 205.3 | 30.8 | 114.0 | 17.1 | 32.5 | 0.4 | 0 | 0 | 0.4 | 154 | 110 |
| SO286_75_15 | 10–15 | 433.8 | 361.1 | 72.7 | 246.4 | 49.6 | 65.3 | 0.3 | 0 | 0 | 0.3 | 157 | 139 |
| SO286_75_20 | 15–20 | 527.6 | 456.4 | 71.2 | 255.1 | 39.8 | 107.1 | 0.3 | 0 | 0 | 0.3 | 159 | 223 |
| SO286_75_25 | 20–25 | 453.4 | 366.1 | 87.3 | 205.1 | 48.9 | 118.2 | 0.2 | 0 | 0 | 0.2 | 155 | 447 |

**Table A3.** List of modern benthic foraminiferal species recognized in samples from six stations of RV Sonne cruise SO286 in abyssal plains of the Labrador Sea, Labrador Basin and southwest of the Azores. * = not figured.

| | |
|---|---|
| *Abditodentrix pseudothalmanni* (Boltovskoy & Guissani de Kahn, 1981) | *Eggerella bradyi* (Cushman, 1911) |
| *Adercotryma glomeratum* (Brady, 1878) | *Epistominella exigua* (Brady, 1884) |
| *Alabaminella weddellensis* (Earland, 1936) | *Eponides repandus* (Fichtel & Moll, 1798) |
| *Ammobaculites agglutinans* (d'Orbigny, 1846) | *Eratidus foliaceus* (Brady, 1881) |
| *Ammobaculites crassaformis* Zheng, 1988 | *Eubuliminella exilis* (Brady, 1884) |
| *Ammobaculites filiformis* Earland, 1934 | *Eusphaeroidina inflata* Ujiié, 1990 |
| *Ammobaculites* ? sp. | *Favulina hexagona* (Williamson, 1848) |
| *Ammodiscus* sp. * | *Fissurina castanea* (Flint, 1899) |
| *Ammoglobigerina globulosa* (Cushman, 1920) | *Fissurina granifera trimarginata* (Buchner, 1940) |
| *Ammolagena clavata* (Jones & Parker, 1860) | *Fissurina orbignyana* var. *rhumbleri* Buchner, 1940 |

**Table A3.** *Cont.*

| | |
|---|---|
| *Ammomassilina alveoliniformis* (Millett, 1898) | *Fissurina staphyllearia* Schwager, 1866 |
| *Anomalinoides globulosus* (Chapman & Parr, 1937) | *Fissurina* sp. 1 |
| *Aschemonella scabra* Brady, 1879 | *Fissurina* sp. 2 * |
| *Bolivina* sp. * | *Fissurina* sp. 3 * |
| *Bulimina buchiana* d'Orbigny, 1846 | *Fissurina* sp. 4 * |
| *Buzasina galeata* (Brady, 1881) | *Fursenkoina texturata* (Brady, 1884) |
| *Buzasina ringens* (Brady, 1879) | *Galwayella trigonoornata* (Brady, 1881) |
| *Cassidulina reniforme* Nørvang, 1945 | *Gavelinopsis* sp. |
| *Cassidulina* sp. * | *Glaphyrammina americana* (Cushman, 1910) |
| *Chilostomella oolina* Schwager, 1878 | *Globocassidulina subglobosa* (Brady, 1881) |
| *Cibicides pachyderma* (Rzehak, 1886) | *Glomospira charoides* (Jones & Parker, 1860) |
| *Cibicides refulgens* Montfort, 1808 | *Glomospira gordialis* (Jones & Parker, 1860) |
| *Cibicidoides cicatricosus* (Schwager, 1866) | *Guttulina communis* (d'Orbigny, 1826) |
| *Cibicidoides mundulus* (Brady, Parker & Jones, 1888) | *Gyroidina* sp. 1 |
| *Cibicidoides wuellerstorfi* (Schwager, 1866) | *Gyroidina* sp. 2 |
| *Cibicidoides* sp. 1 * | *Hansenisca soldanii* (d'Orbigny, 1826) |
| *Cibicidoides* sp. 2 * | *Hoeglundina elegans* (d'Orbigny, 1826) |
| *Cibicidoides* sp. 3 * | *Hormosinelloides guttifer* (Brady, 1884) |
| *Cibicidoides* sp. 4 * | *Hyalinea balthica* (Schröter, 1783) |
| *Cornuloculina inconstans* (Brady, 1879) | *Hyperammina elongata* Brady, 1878 |
| *Cornuspira carinata* (Costa, 1856) | *Ioanella tumidula* (Brady, 1884) |
| *Cribrostomoides jeffreysii* (Williamson, 1858) | *Karreriella bradyi* (Cushman, 1911) |
| *Cribrostomoides sphaerilocula* (Cushman, 1910) | *Karrerulina conversa* (Grzybowski, 1901) |
| *Cribrostomoides subglobosus* (Cushman, 1910) | *Laevidentalina haueri* (Neugeboren, 1856) |
| *Cribrostomoides* sp. * | *Laevidentalina* sp. * |
| *Cystammina pauciloculata* (Brady, 1879) | *Lagena striata* (d'Orbigny, 1839) |
| *Discorbinella complanata* (Sidebottom, 1918) | *Lagena sulcata* (Walker & Jacob, 1798) |
| *Lagena wiesneri* Parr, 1950 | *Pyrulina angusta* (Egger, 1857) |
| *Lagena* sp. 1 | *Pyrulina cylindroides* (Roemer, 1838) |
| *Lagena* sp. 2 | *Pyrulina fusiformis* (Roemer, 1838) |
| *Lagena* sp. 3 * | *Pyrulina* sp. * |
| *Lagena* sp. 4 * | *Quinqueloculina venusta* Karrer, 1868 |
| *Lagenammina arenulata* (Skinner, 1961) | *Quinqueloculina vulgaris* d'Orbigny, 1826 |
| *Lagenosolenia incomposita* Patterson & Pettis, 1986 | *Quinqueloculina* sp. 1 * |
| *Lagnea radiata* (Seguenza, 1862) | *Quinqueloculina* sp. 2 * |
| *Laticarinina pauperata* (Parker & Jones, 1865) | *Quinqueloculina* sp. 3 * |
| *Lenticulina convergens* (Bornemann, 1855) | *Quinqueloculina* sp. 4 * |
| *Lenticulina* sp. 1 * | *Recurvoides contortus* Earland, 1934 |
| *Lenticulina* sp. 2 * | *Reophax scorpiurus* de Montfort, 1808 |
| *Lobatula lobatula* (Walker & Jacob, 1798) | *Reophax* sp. 1 |
| *Melonis affinis* (Reuss, 1851) | *Reophax* sp. 2 * |

**Table A3.** *Cont.*

| | |
|---|---|
| *Melonis pompiloides* (Fichtel & Moll, 1798) | *Reophax* sp. 3 |
| *Miliolinella subrotunda* (Montagu, 1803) | *Rhizammina algaeformis* Brady, 1879 |
| *Nonionellina labradorica* (Dawson, 1860) | *Robertinoides bradyi* (Cushman & Parker, 1936) |
| *Nuttallides decorata* (Phleger & Parker, 1951) | *Rutherfordoides rotundatus* (Parr, 1950) |
| *Oolina globosa* (Montagu, 1803) | *Rutherfordoides rotundiformis* (McCulloch, 1977) |
| *Oridorsalis umbonatus* (Reuss, 1851) | *Saccorhiza ramosa* (Brady, 1879) |
| *Oridorsalis* sp. * | *Sigmoilopsis schlumbergeri* (Silvestri, 1904) |
| *Patellina corrugata* Williamson, 1858 | *Siphotextularia rolshauseni* Phleger & Parker, 1951 |
| *Patellina simplissima* (McCulloch, 1977) | *Sphaeroidina bulloides* d'Orbigny in Deshayes, 1828 |
| *Placopsilinella aurantiaca* Earland, 1934 | *Spiroloculina excavata* d'Orbigny, 1846 |
| *Portatrochammina* sp. | *Spirophthalmidium acutimargo* (Brady, 1884) |
| *Procerolagena gracilis* (Williamson, 1848) | *Spirosigmoilina pusilla* (Earland, 1934) |
| *Protoglobobulimina* sp. | *Subreophax aduncus* (Brady, 1882) |
| *Psammosphaera fusca* Schulze, 1875 | *Tolypammina schaudinni* Rhumbler, 1904 |
| *Pseudononion granuloumbilicatum* Zheng, 1979 | *Triloculina oblonga* (Montagu, 1803) |
| *Pseudopolymorphina novangliae* (Cushman, 1923) | *Triloculina trihedra* Loeblich & Tappan, 1953 |
| *Pullenia bulloides* (d'Orbigny, 1826) | *Triloculina* sp. * |
| *Pullenia quinqueloba* (Reuss, 1851) | *Tritaxis heronalleni* Mikhalevich, 1972 |
| *Pyrgo lucernula* (Schwager, 1866) | *Trochammina* sp. 1 * |
| *Pyrgo murrhina* (Schwager, 1866) | *Trochammina* sp. 2 * |
| *Pyrgo simplex* (d'Orbigny, 1846) | *Uvigerina* sp. 1 |
| *Pyrgo williamsoni* (Silvestri, 1923) | *Uvigerina* sp. 2 |
| *Pyrgo* sp. * | *Uvigerina* sp. 3 * |
| *Pyrgoella* sp. | *Agglutinated tube not assigned to the order Astrorhizida* |

**Table A4.** Diversity measures of the clusters I–IV in samples from six stations of RV Sonne cruise SO286 in abyssal plains of the Labrador Sea, Labrador Basin and southwest of the Azores.

| Sample | Cluster | Number of Taxa | Fisher's Alpha ($\alpha$) | | Equitability (J) | | Shannon (H) | |
|---|---|---|---|---|---|---|---|---|
| | | | Single | Mean | Single | Mean | Single | Mean |
| SO286_21_02 | I | 40 | 17.05 | | 0.93 | | 3.44 | |
| SO286_21_04 | I | 41 | 18.35 | | 0.86 | | 3.20 | |
| SO286_21_06 | I | 39 | 16.97 | | 0.88 | | 3.23 | |
| SO286_21_08 | I | 36 | 14.40 | 14.60 | 0.81 | 0.83 | 2.92 | 2.98 |
| SO286_21_10 | I | 34 | 13.55 | | 0.80 | | 2.81 | |
| SO286_21_15 | I | 31 | 11.35 | | 0.78 | | 2.69 | |
| SO286_21_20 | I | 29 | 10.52 | | 0.77 | | 2.59 | |
| SO286_42_02 | II | 35 | 13.93 | | 0.77 | | 2.74 | |
| SO286_42_04 | II | 37 | 15.10 | | 0.63 | | 2.28 | |
| SO286_42_06 | II | 40 | 17.33 | | 0.74 | | 2.73 | |
| SO286_42_08 | II | 29 | 10.60 | | 0.76 | | 2.57 | |
| SO286_42_10 | II | 33 | 12.84 | 14.14 | 0.71 | 0.73 | 2.47 | 2.60 |
| SO286_42_15 | II | 36 | 14.78 | | 0.75 | | 2.70 | |
| SO286_42_20 | II | 33 | 12.70 | | 0.70 | | 2.45 | |
| SO286_42_25 | II | 38 | 15.82 | | 0.79 | | 2.86 | |

**Table A4.** *Cont.*

| Sample | Cluster | Number of Taxa | Fisher's Alpha (α) | | Equitability (J) | | Shannon (H) | |
|---|---|---|---|---|---|---|---|---|
| | | | Single | Mean | Single | Mean | Single | Mean |
| SO286_42_30 | III | 40 | 17.69 | | 0.88 | | 3.24 | |
| SO286_65_09 | III | 42 | 19.19 | | 0.81 | | 3.01 | |
| SO286_65_11 | III | 45 | 21.38 | | 0.84 | | 3.19 | |
| SO286_66 | III | 38 | 16.00 | 23.26 | 0.85 | 0.87 | 3.10 | 3.37 |
| SO286_67_02 | III | 48 | 23.36 | | 0.87 | | 3.36 | |
| SO286_67_04 | III | 55 | 30.10 | | 0.90 | | 3.61 | |
| SO286_67_06 | III | 50 | 25.72 | | 0.88 | | 3.44 | |
| SO286_67_08 | III | 49 | 25.32 | | 0.93 | | 3.62 | |
| SO286_67_10 | III | 48 | 23.58 | | 0.89 | | 3.45 | |
| SO286_75_02 | IV | 49 | 24.81 | | 0.90 | | 3.49 | |
| SO286_75_04 | IV | 58 | 33.84 | | 0.92 | | 3.72 | |
| SO286_75_06 | IV | 49 | 25.19 | | 0.93 | | 3.62 | |
| SO286_75_08 | IV | 49 | 24.21 | 25.86 | 0.93 | 0.92 | 3.62 | 3.62 |
| SO286_75_10 | IV | 48 | 23.92 | | 0.92 | | 3.57 | |
| SO286_75_15 | IV | 47 | 22.73 | | 0.93 | | 3.60 | |
| SO286_75_20 | IV | 50 | 25.09 | | 0.94 | | 3.68 | |
| SO286_75_25 | III | 49 | 24.69 | ^ | 0.91 | ^ | 3.52 | ^ |

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
