# Peer review of "Benthic Foraminifera Diversity of the Abyssal Northwest Atlantic"

_diversity, doi:10.3390/d15030381_

Round 1
Reviewer 1 Report
This study describes Recent benthic foraminiferal assemblages in core samples collected from abyssal plains in the North Atlantic during RV Sonne cruise SO286 and illustrates species found in the four faunas identified by cluster analysis.
- Although I requested the assistant editor to send me Supplementary materials, Tables S1 and S2, twice, I never received them. Therefore, I refrain from making detailed comments/suggestions on the results of cluster analysis and interpretation as I cannot verify without access to the raw data.
- I agree with most species identification, except for some specimens that clearly show morphological features not typical of a genus or species (please see comments in the PDF). For Rotaliid foraminifera, it is always better to provide the illustration of three sides (dorsal, ventral, and peripheral) of a specimen, especially if the purpose of this manuscript is the documentation of diversity. Although I said I agree with the identification, with the illustration of two sides, for some specimens, it is not possible to verify the species identification and suggest an alternative taxon.
- The quality of the illustration of specimens in the 12 plates is excellent as expected from the author. However, I suggest re-positioning figure alphabets and scales.
- The manuscript needs some figures to illustrate the changes in assemblages that the author describes in the text and to support the author's arguments because the majority of the data are in the appendix and the supplementary materials (please see my comment in page 10 of the PDF).
- The manuscript will benefit from adding more oceanographic and environmental information collected during the cruise, for example, to explain differences in the composition of the four faunas.
- I suggest having someone who can help you edit the text and tables. Some paragraphs are redundant. Tables, especially Table A2 with many abbreviations, are hard to understand. They can be formatted with different line types/thicknesses and shades to make them more reader-friendly.
- For other comments, please see the annotations in the PDF.

Author Response
This study describes Recent benthic foraminiferal assemblages in core samples collected from abyssal plains in the North Atlantic during RV Sonne cruise SO286 and illustrates species found in the four faunas identified by cluster analysis.
This study describes Recent benthic foraminiferal assemblages in core samples collected from abyssal plains in the North Atlantic during RV Sonne cruise SO286 and illustrates species found in the four faunas identified by cluster analysis.
- Although I requested the assistant editor to send me Supplementary materials, Tables S1 and S2, twice, I never received them. Therefore, I refrain from making detailed comments/suggestions on the results of cluster analysis and interpretation as I cannot verify without access to the raw data.
=> Editor: if the reviewer does not get the full paper she/he cannot make a full review
- I agree with most species identification, except for some specimens that clearly show morphological features not typical of a genus or species (please see comments in the PDF). For Rotaliid foraminifera, it is always better to provide the illustration of three sides (dorsal, ventral, and peripheral) of a specimen, especially if the purpose of this manuscript is the documentation of diversity. Although I said I agree with the identification, with the illustration of two sides, for some specimens, it is not possible to verify the species identification and suggest an alternative taxon.
=> was dealt with in the PDF comments
- The quality of the illustration of specimens in the 12 plates is excellent as expected from the author. However, I suggest re-positioning figure alphabets and scales.
=> adjusted on all plates
- The manuscript needs some figures to illustrate the changes in assemblages that the author describes in the text and to support the author's arguments because the majority of the data are in the appendix and the supplementary materials (please see my comment in page 10 of the PDF).
=> the data are given in the appendix and the supplementary materials
- The manuscript will benefit from adding more oceanographic and environmental information collected during the cruise, for example, to explain differences in the composition of the four faunas.
=> such data were not available and beyond the scope of the paper. The author chose not to deal with oceanographic and environmental information as material from only six sites taken at one date was available. Conclusions would have been based on very few singular data points.
- I suggest having someone who can help you edit the text and tables. Some paragraphs are redundant. Tables, especially Table A2 with many abbreviations, are hard to understand. They can be formatted with different line types/thicknesses and shades to make them more reader-friendly.
=> grey shaded areas in the tables have been added, the rest will be dealt with on concrete annotations in the PDF. Some redundant sentences where erased.
- For other comments, please see the annotations in the PDF.
dealt with as described below
Comments on the suggestions in the pdf.
Line 11 => changed to “not assigned to a species”
Line 20 => The text has been adjusted. As the summed up share of rare species is not rare, they should not be excluded. It is the major conclusion of the paper that the rare species give a signal, which should not be ignored.
Line 22 important was replaced by “not rare to dominant”
Line 42 some samples are from the flank of a seamount and the caption of table 2 was adjusted
Line 78 “geographically” was added
Line 101 Editor: if the reviewer does not get the full paper she/he cannot make a full review
Line 273 The interpretation is based on the sedimentation rate and this sentence was moved to the end.
Line 283-287 The data are given in table A4 of the appendix and are easy to follow there.
Plates: Figure alphabets have been repositioned on all plates
Plate A1 B, O are not identified as Reophax sp. 3 // C is identified as “Agglutinated tube not assigned to a taxon” // E: No specimen with a proloculus was found.
Plate A2 F: yes it is evolute. // K: changed to Karrerulina conversa. //L: changed to Ammobaculites ? sp.
Plate A7 F: new image and changed to Protoglobobulimina sp. similar to Jones, Robert Wynn, 1994: The Challenger Foraminifera Oxford Univ Press 416 pp. Plate 50, Fig. 14
Plate A7 H (now I) changed to Uvigerina sp. 2
Plate A10 H: changed to Gyroidina sp. 1
Due to the changes in identifications numbers in the text and tables needed to be adjusted
Reviewer 2 Report
line 9: "is documented for the first time."
line 128: Mention the diameter of the MUC tube
line 134 & 291: Ecologically, density or standing crop refers to specimens per unit area or volume (not usually mass). If you know the diameter of your MUC tube you can calculate this out.
line 158: Add "Rarefaction curve" to figure caption
line 217: Spelling of Hyperammina
Line 309: Cite Gooday, 1988
line 484: I think this photo may be Cibicidoides bradyi considering the size and the ventral arrangement of chambers
line 346: "understudied" sounds better than little investigated.
Line 439: Spelling of "Lagena"
Mention that this study represents Holocene and Pleistocene assemblages in the abstract or even title. Should be advertised rather than just mentioned once in the discussion.
Methodology question: How were tube species quantified? Did you count fragments as whole individuals?
Micrographs are great!
Author Response
line 9: "is documented for the first time." => adjusted accordingly
line 128: Mention the diameter of the MUC tube => adjusted accordingly
line 134 & 291: Ecologically, density or standing crop refers to specimens per unit area or volume (not usually mass). If you know the diameter of your MUC tube you can calculate this out.
=> the exact volume could not be determined due to loss of material while separating and packing on board so the mass was used instead.
line 158: Add "Rarefaction curve" to figure caption => adjusted accordingly
line 217: Spelling of Hyperammina => adjusted accordingly
Line 309: Cite Gooday, 1988 => adjusted accordingly
line 484: I think this photo may be Cibicidoides bradyi considering the size and the ventral arrangement of chambers => no change was made
line 346: "understudied" sounds better than little investigated. => adjusted accordingly
Line 439: Spelling of "Lagena" => According to the WFD Lagnea radiata is the valid name and Lagnea a genus with several valid species.
Mention that this study represents Holocene and Pleistocene assemblages in the abstract or even title. Should be advertised rather than just mentioned once in the discussion. => an according sentence was added to the abstract
Methodology question: How were tube species quantified? Did you count fragments as whole individuals? => short tube fragments were very scarce and not counted.
Micrographs are great!
Reviewer 3 Report
Dear Michael Hesemann,
thank you for this nicely organized manuscript and the detailed plates on foraminiferan species. This is a very nice visualization and appreciated input to the North Atlantic benthic community. Please indicate species, when mentioning them the first time, taxonomically correct with author and year. Most important is a more clear "red line" in discussion and conclusion. The results are very nice, but could be imbedded into the project goals with a bit mire detail (see last comment below).
I have only a few minor comments:
Abstract: p1, line 10 ... Southwest of the Azores is for the first time documented." Please delete "for the first time". That you submitted this paper indicated novelty already.
p1, line 21"Epistominella exigua" Please add author and year when you mention a species a first time. Please check this throughout the whole manuscript for all first time mentions of a taxon (species).
Introduction p2 line 57 "Foraminifera are placed as a phylum into the infra-kingdom Rhizaria." I would delete this sentence. It is not logically as link between the content of fossils. The sentence before ends with fossils and the sentence after continues.
Results, p10, line 300 Italic species name in an italic heading is may be not the best choice.
Conclusion p11 and p 12. This is more a summary of your paper again instead of a conclusion. I would have liked to see an anwer on the question what the formainiferan assemblages in the four different clusters may mean for the benthic species in these areas feeding on foraminiferans? Would there be a context for sediment structure and foraminiferan community? Are the species known as food source for other benthic organisms? You are speaking about season ifluencing the samping, but can you go for more detail here, what this would mean for the benthic community as "snapshot" of the sampling? Well, the conclusion can be a bit more open to some aspects in the discussion.
Author Response
Abstract: p1, line 10 ... Southwest of the Azores is for the first time documented." Please delete "for the first time". That you submitted this paper indicated novelty already.
=> adjusted accordingly
p1, line 21"Epistominella exigua" Please add author and year when you mention a species a first time. Please check this throughout the whole manuscript for all first time mentions of a taxon (species).
=> adjusted accordingly in the text, though it is not common practice in papers of the journal Diversity
Introduction p2 line 57 "Foraminifera are placed as a phylum into the infra-kingdom Rhizaria." I would delete this sentence. It is not logically as link between the content of fossils. The sentence before ends with fossils and the sentence after continues.
=> sentence deleted
Results, p10, line 300 Italic species name in an italic heading is may be not the best choice.
=> There is no alternative
Conclusion p11 and p 12. This is more a summary of your paper again instead of a conclusion. I would have liked to see an anwer on the question what the formainiferan assemblages in the four different clusters may mean for the benthic species in these areas feeding on foraminiferans? Would there be a context for sediment structure and foraminiferan community? Are the species known as food source for other benthic organisms? You are speaking about season ifluencing the samping, but can you go for more detail here, what this would mean for the benthic community as "snapshot" of the sampling? Well, the conclusion can be a bit more open to some aspects in the discussion.
=> data on physical, chemical and organic material parameters were not available, so conclusions on feeding, sediment structure or predators could not be drawn. To repeat what is known or assumed from other papers from abyssal areas would not have added new information and it would have been too speculative to apply it to the sample locality.